# Photoreceptor loss does not recruit neutrophils despite strong microglial activation

Derek Power[1,2], Justin Elstrott[3], Jesse Schallek[1,2,4]*

[1]Center for Visual Science, University of Rochester, Rochester, United States; [2]Flaum Eye Institute, University of Rochester, Rochester, United States; [3]Department of Translational Imaging, Genentech, Inc, South San Francisco, United States; [4]Department of Neuroscience, University of Rochester, Rochester, United States

## eLife Assessment

The study by Power and colleagues is **important** as elucidating the dynamic immune responses to photoreceptor damage in vivo potentiates future work in the field to better understand the disease process. The evidence supporting the authors' claims is **compelling**.

*For correspondence:
jschall3@ur.rochester.edu

**Abstract** In response to central nervous system (CNS) injury, tissue-resident immune cells such as microglia and circulating systemic neutrophils are often first responders. The degree to which these cells interact in response to CNS damage is poorly understood, and even less so, in the neural retina, which poses a challenge for high-resolution imaging in vivo. In this study, we deploy fluorescence adaptive optics scanning light ophthalmoscopy (AOSLO) to study microglia and neutrophils in mice. We simultaneously track immune cell dynamics using label-free phase-contrast AOSLO at micron-level resolution. Retinal lesions were induced with 488 nm light focused onto photoreceptor (PR) outer segments. These lesions focally ablated PRs, with minimal collateral damage to cells above and below the plane of focus. We used in vivo AOSLO, and optical coherence tomography (OCT) imaging to reveal the natural history of the microglial and neutrophil response from minutes to months after injury. While microglia showed dynamic and progressive immune response with cells migrating into the injury locus within 1 day after injury, neutrophils were not recruited despite close proximity to vessels carrying neutrophils only microns away. Post-mortem confocal microscopy confirmed in vivo findings. This work illustrates that microglial activation does not recruit neutrophils in response to acute, focal loss of PRs, a condition encountered in many retinal diseases.

## Introduction

In the mammalian retina, a rapid and coordinated immune response to infection or injury is important for maintaining tissue homeostasis. This is especially critical in the eye since mature retinal neurons do not typically regenerate, resulting in long-term functional losses for the host (*Stone et al., 1999*; *García-Ayuso et al., 2019*). The retina is considered immune-privileged and is equipped with a resident population of innate immune cells, including microglia (*Boycott and Hopkins, 1981*; *Silverman and Wong, 2018*). In healthy retina, microglia are distributed primarily in the inner retina, residing within nerve fiber layer (NFL), inner plexiform layer (IPL), and outer plexiform layer (OPL) (*Wang et al., 2016*; *Zhang et al., 2018*), generally avoiding the nuclear layers. Microglia tile the retina and, like their counterparts in the brain, exhibit long, thin processes that continually probe the neuro-glial microenvironment (*Silverman and Wong, 2018*; *Joseph et al., 2021*; *Nimmerjahn et al., 2005*).

In addition to phagocytosing debris (*Neumann et al., 2009*; *Zabel et al., 2016*), regulating synaptic maintenance (*Tremblay and Majewska, 2011*; *Sipe et al., 2016*), and removing dead tissue (*Márquez-Ropero et al., 2020*; *Karlen et al., 2020*), microglia can secrete chemokines to recruit other leukocytes to help fight infection and repair damaged tissue (*Silverman and Wong, 2018*; *Okunuki et al., 2019*; *Kremlev et al., 2004*; *Babcock et al., 2003*). For many injuries, one of the first systemic responders recruited and activated by microglia are neutrophils (*Boyce et al., 2022*; *Zhou et al., 2006*). Neutrophils comprise a large fraction (20–30%) of leukocytes in murine blood (*Provencher et al., 2010*). They assist in maintaining tissue homeostasis by neutralizing foreign agents, regulating the immune response, and phagocytosing dead tissue (*Burn et al., 2021*; *Peiseler and Kubes, 2019*). Under inflammatory conditions, the spatiotemporal interplay between microglia and neutrophils is poorly understood. A missed window of interaction is highly problematic in histological study where a single time point reveals a snapshot of the temporally complex immune response, which changes dynamically over time. Here, we use in vivo imaging to overcome these constraints.

Documenting immune cell interactions in the retina over time has been challenged by insufficient resolution and contrast to visualize single cells in the living eye. The microscopic size of immune cells requires exceptional resolution for detection. Recently, advances in adaptive optics scanning light ophthalmoscopy (AOSLO) imaging have provided micron-level resolution and enhanced contrast for imaging individual immune cells in the retina without requiring extrinsic dyes (*Joseph et al., 2021*; *Joseph et al., 2020*). AOSLO provides multimodal information from confocal reflectance, phase-contrast, and fluorescence modalities, which can reveal a variety of cell types simultaneously in the living eye. Here, we used confocal AOSLO to track changes in reflectance at cellular scale. Phase-contrast AOSLO provides detail on highly translucent retinal structures such as vascular wall, single blood cells (*Guevara-Torres et al., 2016*; *Joseph et al., 2019a*; *Joseph et al., 2019b*), photoreceptor (PR) somata (*Guevara-Torres et al., 2015*), and is well-suited to image resident and systemic immune cells (*Joseph et al., 2021*; *Joseph et al., 2020*). Fluorescence AOSLO provides the ability to study fluorescently labeled cells (*Guevara-Torres et al., 2020*; *Geng et al., 2012*; *Schallek et al., 2013*) and exogenous dyes (*Guevara-Torres et al., 2016*; *Pinhas et al., 2013*) throughout the living retina. These modalities used in combination have recently provided detailed images of the retinal response

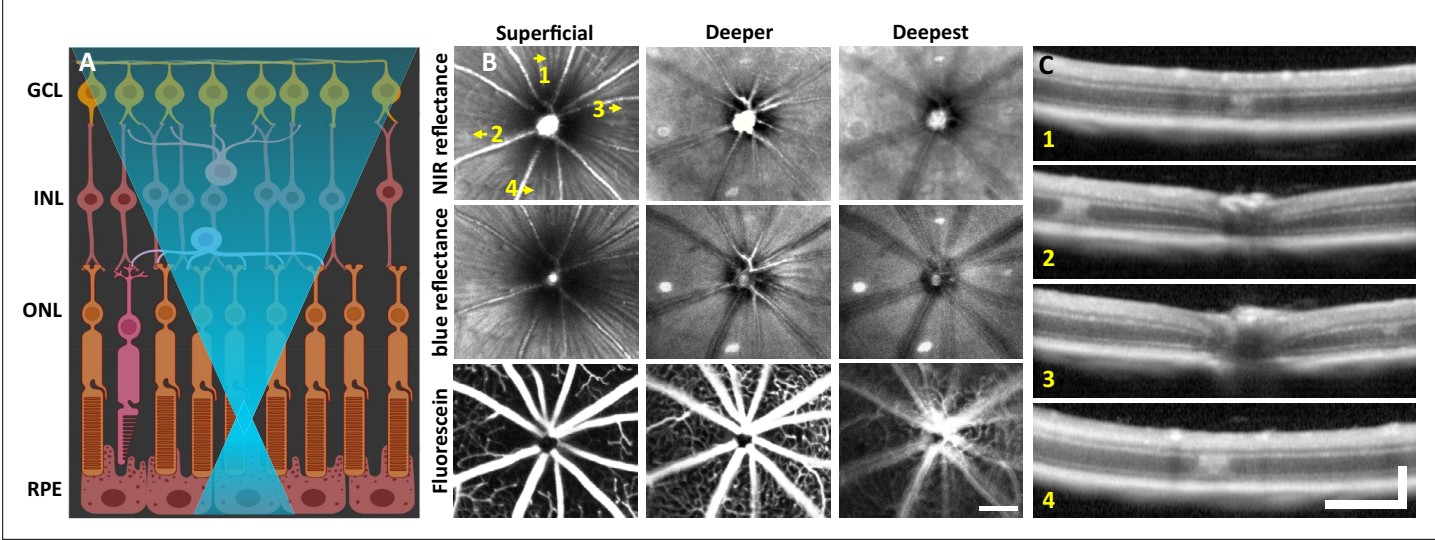

**Figure 1.** Laser injury assessed with commercial scanning light ophthalmoscopy (SLO) and optical coherence tomography (OCT). (**A**) 488 nm light is focused onto the photoreceptor outer segments using adaptive optics scanning light ophthalmoscopy (AOSLO). Created with BioRender.com. (**B**) 30° SLO images of near-infrared (NIR) reflectance, blue reflectance, and fluorescein angiography of a mouse retina 1 day after laser exposure. Three focal planes are shown. NIR and blue reflectance reveal small hyperreflective regions below the superficial plane. Fluorescein reveals intact vasculature with no sign of leakage. Arrows indicate regions with imparted laser damage (1–4). (**C**) OCT B-scans passing through laser-exposed regions indicated in (**B**). Exposures produced a focal hyperreflective band within the outer nuclear layer (ONL) with adjacent retina appearing healthy. OCT images were spatially averaged (~30 µm, three B-scans). Scale bars = 200 µm horizontal, 200 µm vertical.

The online version of this article includes the following figure supplement(s) for figure 1:

**Figure supplement 1.** Lesion location tracked from minutes to 1 day with optical coherence tomography (OCT).

to a model of human uveitis (*Joseph et al., 2020*; *Chu et al., 2016*). Together, these innovations now provide a platform to visualize, for the first time, the dynamic interplay between many immune cell types, each with a unique role in tissue inflammation. We combine these innovative modalities with conventional histology and commercial scanning light ophthalmoscopy (SLO)/optical coherence tomography (OCT) to reveal the progressive nature of the cellular response to acute retinal injury.

Here, we ask the question: 'To what extent do microglia/neutrophils respond to acute neural loss in the retina?' To begin unraveling the complexities in this response, we deploy a deep retinal laser ablation model. Using AOSLO, we track and characterize the changes in microglia, neutrophils, and retinal structure within hours, days, and months after acute laser exposure.

## Results

### Characterization of deep focal laser damage

Four complementary imaging modalities were used to evaluate the nature and localization of the focal laser damage induced by 488 nm light: wide field SLO, OCT, AOSLO, and post-mortem histology. Each is reported in turn below.

### Laser damage induces focal hyperreflective lesions in outer retina imaged with wide-field SLO

To observe global and focal retinal health, we used commercial SLO. 1 day post-488 nm light exposure, both near-infrared (NIR) and blue reflectance modalities showed hyperreflective lesions, most apparent at a deeper retinal focus position. In the inner retina, lesions were not visible and the retina appeared healthy (*Figure 1B*). Despite deep retinal damage, fluorescein angiography did not reveal dye leakage (*Figure 1B*, bottom), indicating the blood–retinal barrier remained intact.

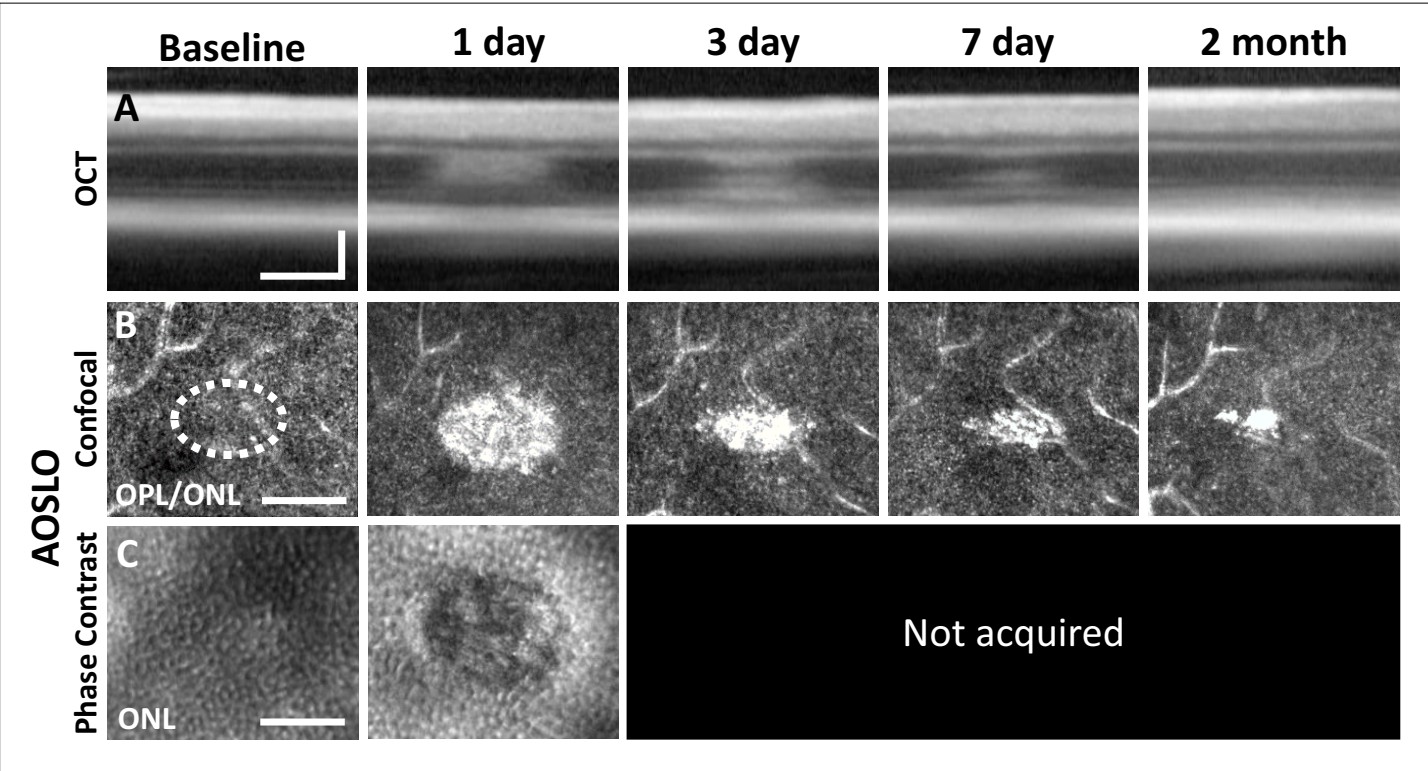

**Figure 2.** Laser damage temporally tracked with adaptive optics scanning light ophthalmoscopy (AOSLO) and optical coherence tomography (OCT). Laser-exposed retina was tracked with OCT (**A**), confocal (**B**), and phase-contrast (**C**) AOSLO for baseline, 1-, 3-, 7-day, and 2-month time points. OCT and confocal AOSLO display a hyperreflective phenotype that was largest/brightest at 1 day and became nearly invisible by 2 months. Dashed oval indicates region targeted for laser injury. Phase-contrast AOSLO revealed disrupted photoreceptor soma 1 day after laser injury. Phase-contrast data was not acquired for remaining time points due to the development of cataract, which obscured the phase-contrast signal. OCT images were spatially averaged (~30 μm, eight B-scans). Scale bars = 40 μm horizontal, 100 μm vertical.

## OCT B-scans reveal outer retinal hyperreflection without inner retinal damage

To assist in determination of which retinal layers are damaged by the 488 nm laser, we used OCT. Within 30 minutes post-laser exposure, OCT revealed a zone of hyperreflection within the ONL (*Figure 1—figure supplement 1*). 1 day post-lesion, the focal hyperreflection remained localized to the ONL (~50 μm wide, *Figure 1C*). Retina outside of damage foci appeared normal. The hyperreflective phenotype persisted through 7 days and was cleared by 2 months (*Figure 2A*). There did not appear to be any cellular excavation, 'cratering,' or evidence of edema for any time point assessed. Bruch's membrane appeared intact for all time points, evidenced by lack of fluorescein leakage at the site of lesion. Additionally, retinal vessels appeared normal in OCT B-scans.

## AOSLO reveals outer retinal damage with confocal and phase-contrast modalities

Confocal AOSLO provided micron-level detail of the lesioned area. Confirming OCT findings, we observed hyperreflective changes localized within the ONL that were, by 1 day, the brightest at the OPL/ONL interface (*Figure 2B*). At this plane, the lesions manifest as an ellipse with the long axis in the direction of the line scan used to create the lesion. The most prominent hyperreflective phenotype was seen at 1 day post-lesion, with diminishing size and brightness by days 3 and 7. The phenotype was largely diminished by 2 months (*Figure 2B*).

Using phase-contrast AOSLO also allowed visualization of translucent cells within the retina, which enabled us to image PR somas of the ONL (*Guevara-Torres et al., 2020*; *Guevara-Torres et al., 2015*). Normally, the ONL is comprised of PR somata that are densely packed (*Guevara-Torres et al., 2015*; *Carter-Dawson and LaVail, 1979*). We found that the dense packing of individual somata was disrupted 1 day post-exposure (~50 μm ovoid, *Figure 2C*), suggesting degradation or ablation of the cell membrane.

## Confirmation of outer retinal cell loss using post-mortem histology

To assess the extent of cell loss caused by focal laser exposure, we performed DAPI staining on whole-mount retinal tissue using the same time points assessed for in vivo imaging. 1 day after laser exposure, we observed mild thickening of the ONL compared to unexposed locations only microns away. At 3 and 7 days, local ONL thinning was observed (*Figure 3A*). En-face planes within the ONL revealed a loss of PR nuclei in the outer aspect of the ONL for 3 and 7 day time points, while inner ONL exhibited little evidence of cell loss (*Figure 3B and C*), illustrating the precise axial confinement of the laser damage induced by this method. By 2 months, the lesion's overall appearance and ONL thickness returned to baseline (*Figure 3A–C*). This histological finding corroborates the OCT findings observed in vivo.

ONL nuclear counts were reduced at locations of laser exposure. In comparison to control locations (501,317 nuclei/mm$^2$±46,198 nuclei/mm$^2$, mean ± SD, similar to previous work; *Jeon et al., 1998*), at 3 and 7 days post-exposure, we found a corresponding reduction in the density of ONL nuclei of 18% (408,965 nuclei/mm$^2$±25,621 nuclei/mm$^2$) and 27% (367,542 nuclei/mm$^2$±9038 nuclei/mm$^2$), respectively. Compared to control locations, the 3- and 7-day data resulted in p-values of 0.17 and 0.07, respectively (Student's paired two-tailed *t*-test). 2 months after damage, PR nuclear densities were indistinguishable from that of control (*Figure 3F*, gray bars). Despite losses of PR nuclei, total cell nuclei within the INL remained unchanged for all time points (*Figure 3F*, black bars). These data indicate that the laser exposure focally ablated PRs whilst leaving inner retinal cells intact.

## Retinal vasculature unaffected by deep retinal lesion

A concern with laser lesions of this type is that it may coagulate retinal vessels. Motion contrast (*Chui et al., 2012*; *Guevara-Torres et al., 2016*) images revealed that the vasculature remained perfused from hours to months after laser damage, suggesting that acute or long-term changes are not imparted by the laser injury (*Figure 4*). None of the primary vascular stratifications within the NFL, IPL, and OPL of the mouse retina (*Schallek et al., 2013*) showed stopped flow as a result of the laser lesion, reinforcing the findings above regarding the axial confinement of the damage to the outer retina.

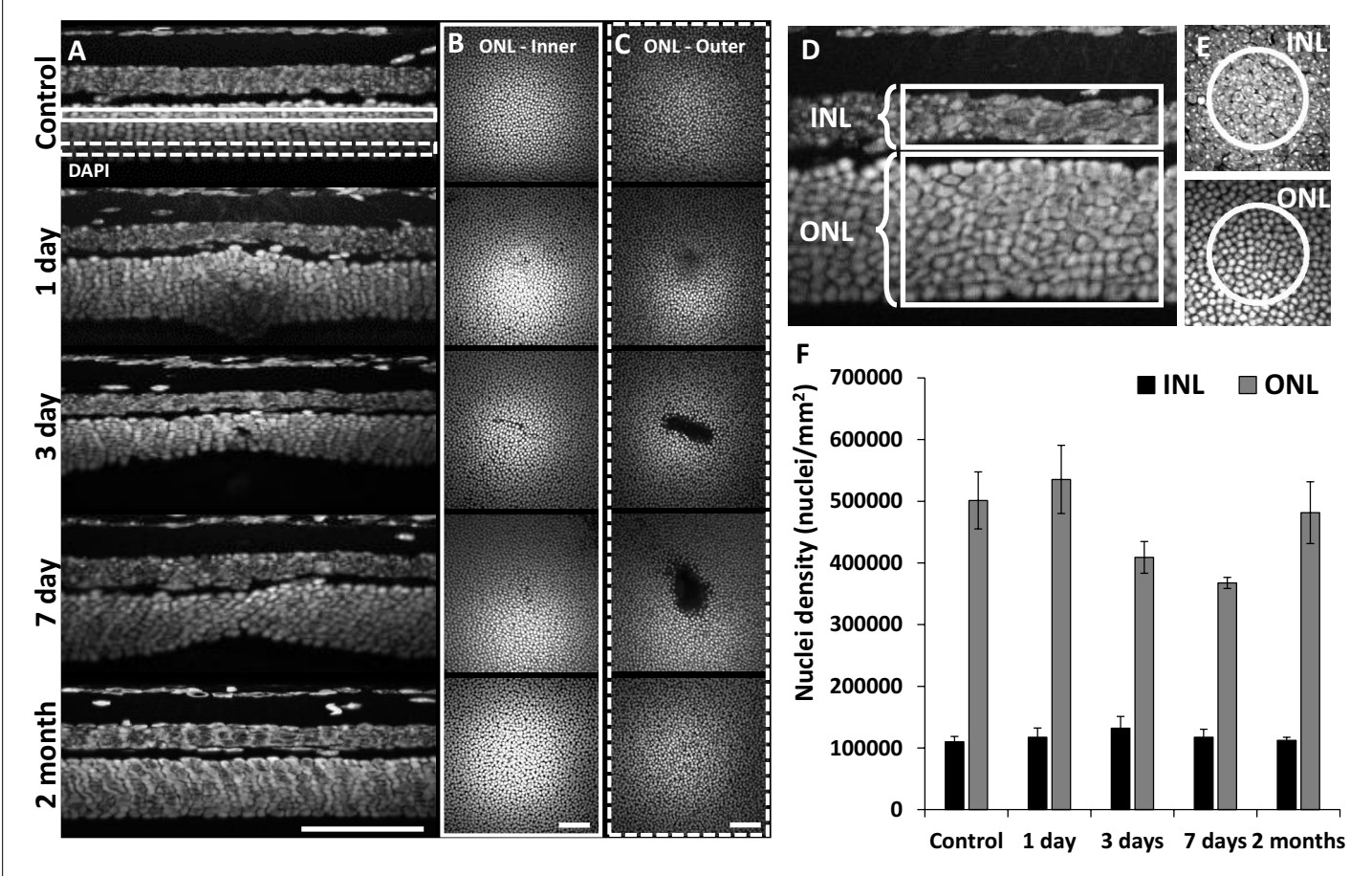

**Figure 3.** Retinal histology confirms photoreceptor ablation and preservation of inner retinal cells. Cross-sectional view (**A**) and en-face (**B, C**) images of DAPI-stained whole-mount retinas at laser injury locations over time. By 1 day, outer nuclear layer (ONL) becomes thicker at the lesion location, but thinner by 3 and 7 days. By 2 months, the ONL appeared similar to that of control. The inner (**B**, solid rectangle) and outer (**C**, dashed rectangle) stratum of ONL show axial differences in ONL loss. Most cell loss was seen in the outer aspect of the ONL (**C**). Scale bars = 40 μm. (**D**) Cross-section of DAPI-stained retina displaying inner nuclear layer (INL) and ONL regions for quantification. Each analysis region was 50 μm across and encompassed the entire depth of the INL or ONL. (**E**) En-face images show 50 μm diameter circles used for analysis. (**F**) Nuclei density for post-injury time points. ONL nuclei were reduced at 3 and 7 days (p=0.17 and 0.07, respectively) while INL density remained stable (n=10 mice, three unique regions per time point). Error bars display mean ± 1 SD.

In addition to perfusion status, phase-contrast AOSLO also permitted analysis of single blood cell flux within capillaries (*Guevara-Torres et al., 2016*; *Dholakia et al., 2022*). We tracked capillary flux at different retinal depths within and above the lesion (IPL, OPL, one capillary each, *Figure 4—figure supplement 1A*). Blood cell flux for capillaries within lesion locations was within the range of normal flux for the C57BL/6 J mouse (*Dholakia et al., 2022*; *Figure 4—figure supplement 1B and C*). Flux tracked from hours to days in these capillaries changed synchronously, displaying positive linear correlation (R²=0.59, *Figure 4—figure supplement 1C and D*). This suggested any such changes in flux were a property of systemic perfusion, rather than locally imparted changes in flow due to the lesion. As an additional control, we evaluated blood flux in two distant capillaries (lesion and control locations, *Figure 4—figure supplement 1E*). Both capillaries displayed similar flux values from minutes to 2 months post-injury (*Figure 4—figure supplement 1F and G*), resulting in positive linear correlation (R²=0.78, *Figure 4—figure supplement 1H*). Taken together, these findings suggest lesions do not appreciably impact local RBC delivery in the capillary network.

## PR laser injury promotes a robust response in nearby microglia

To observe the microglial response to PR laser injury, we imaged fluorescent microglia in CX3CR1-GFP mice with both SLO and AOSLO. 1 day after injury, SLO revealed bright, focal congregations of

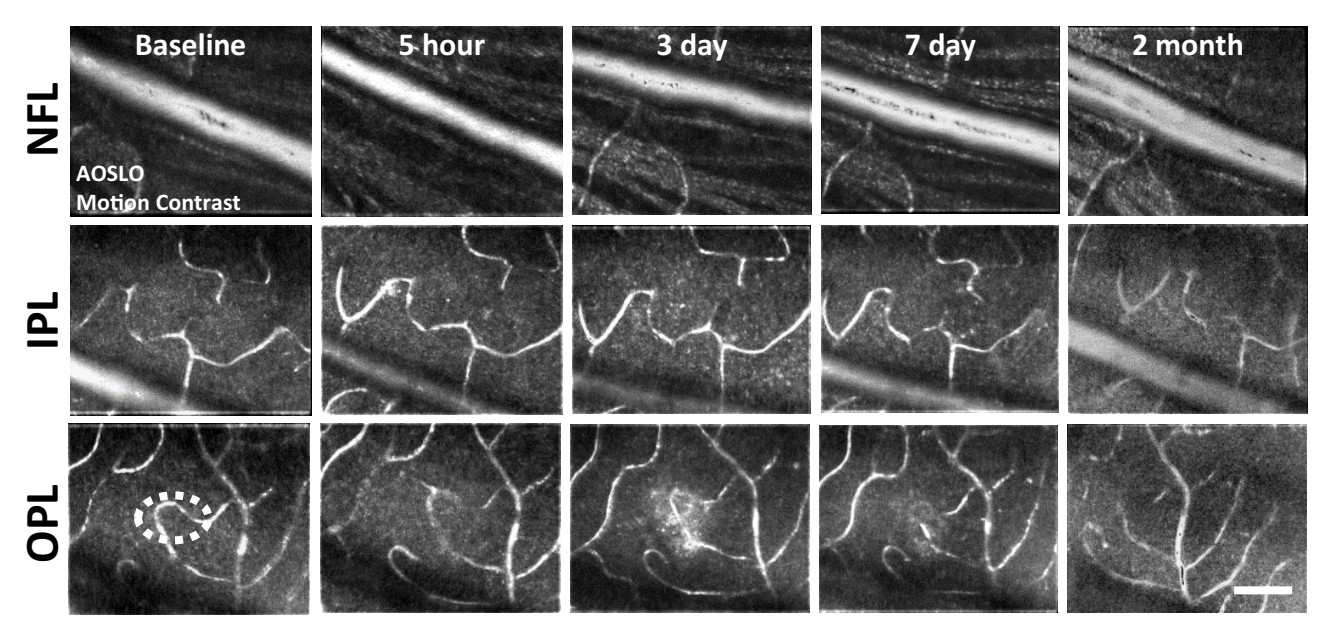

**Figure 4.** Motion-contrast images reveal vascular perfusion status in response to laser damage. A single location was tracked over time at three vascular plexuses using adaptive optics scanning light ophthalmoscopy (AOSLO). Retinal vasculature remained perfused for all time points tracked and at all depths. White oval indicates damage location. Scale bar = 40 μm.

The online version of this article includes the following figure supplement(s) for figure 4:

**Figure supplement 1.** Measurement of single-cell blood flux after laser damage using phase-contrast adaptive optics scanning light ophthalmoscopy (AOSLO).

microglia at injury locations in contrast to undamaged locations, which maintained a distribution of lateral tiling (*Figure 5A and B*). The global visualization of microglia was augmented by high-resolution fluorescence AOSLO, providing enhanced detail of the microglial response to laser injury (*Joseph et al., 2021*; *Wahl et al., 2019*). Whereas AOSLO imaging of microglia in the healthy retina displayed a distributed array of microglia with ramified processes (*Figure 5C*, left), laser-damaged locations showed a congregation of cells 1 day post-injury with less lateral ramification (*Figure 5C*, right). Within hours of the laser exposure, we did not observe a photo-bleaching or death of regional microglia, suggesting that while the laser exposure was sufficient to damage PRs, it left retinal microglia intact.

With phase-contrast imaging targeting the ONL, we documented a rare event of putative pseu-dopod extension at a lesion site (*Figure 5—video 1*). Given the axial complexity of microglia in this layer, it is now possible that microglial process dynamics may be revealed with this label-free approach.

We returned to the same laser-damaged locations to capture microglial appearance at baseline, 1-, 3-, 7-day and 2-month time points with AOSLO to track the natural history of the microglial response to PR damage. At 1 day post-injury, damage locations displayed aggregations of microglia. The surrounding microglia displayed process polarization with extensions projecting toward the injury (*Figure 6*). By days 3 and 7, microglia exhibited fewer lateral projections and somas have migrated into the ONL where they do not normally reside (*Figure 6*). By 2 months post-injury, the hyperreflec-tive phenotype was absent and microglia once again occupied only the inner retina. The microglial distribution at 2 months was similar to baseline, with cells exhibiting radially symmetrical branching projections, similar to those prior to laser injury (*Figure 6*).

A standing question in OCT/confocal AOSLO lesion interpretation is whether microglia contribute or directly produce the hyperreflective phenotype seen in axial B-scans and en-face fundus images (*Puthenparampil et al., 2022*; *Pilotto et al., 2022*). With confocal AOSLO, we find that the hyper-reflective phenotype is visible as early as 30 minutes, becoming larger and brighter by 90 minutes. During these time points, simultaneously imaged microglia remained ramified and maintained a tiled arrangement, indicating that microglia are not the initial source of the lesion-induced hyperreflective appearance (*Figure 6—figure supplement 1*).

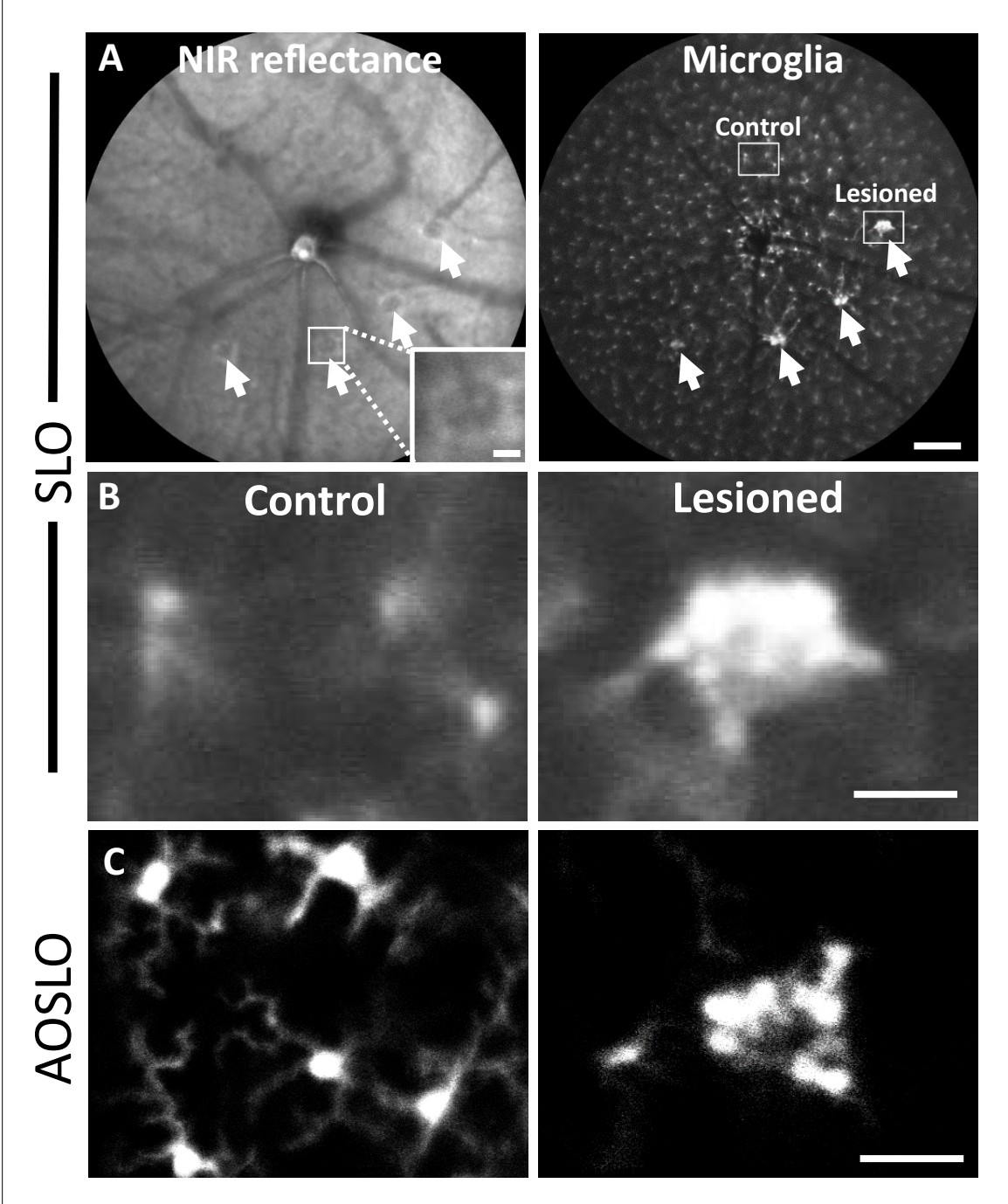

**Figure 5.** Microglial response 1 day after laser injury imaged in vivo with fluorescence scanning light ophthalmoscopy (SLO) and adaptive optics scanning light ophthalmoscopy (AOSLO). (**A**) Left: deep-focus near-infrared (NIR) SLO fundus image (55° FOV) of laser-injured retina. White arrowheads point to damaged locations showing hyperreflective regions. Inset scale bar = 40 μm. Right: fluorescence fundus image from same location. Fluorescent CX3CR1-GFP microglia are distributed across the retina and show congregations at laser-damaged locations. Scale bar = 200 μm. (**B**) Magnified SLO images of microglia at laser-damaged and control locations (indicated in A, right, white boxes). Control location displays distributed microglial, whereas microglia at the lesion location are bright and focally aggregated. (**C**) Fluorescence AOSLO images show greater detail of cell morphology at the same scale. In control locations, microglia showed ramified morphology and distributed concentration, whereas damage locations revealed dense aggregation of many microglia that display less ramification. Scale bars = 40 μm.

The online version of this article includes the following video for figure 5:

**Figure 5—video 1.** Dynamic pseudopodia imaged with phase-contrast adaptive optics scanning light ophthalmoscopy 1 day post-injury.
https://elifesciences.org/articles/98662/figures#fig5video1

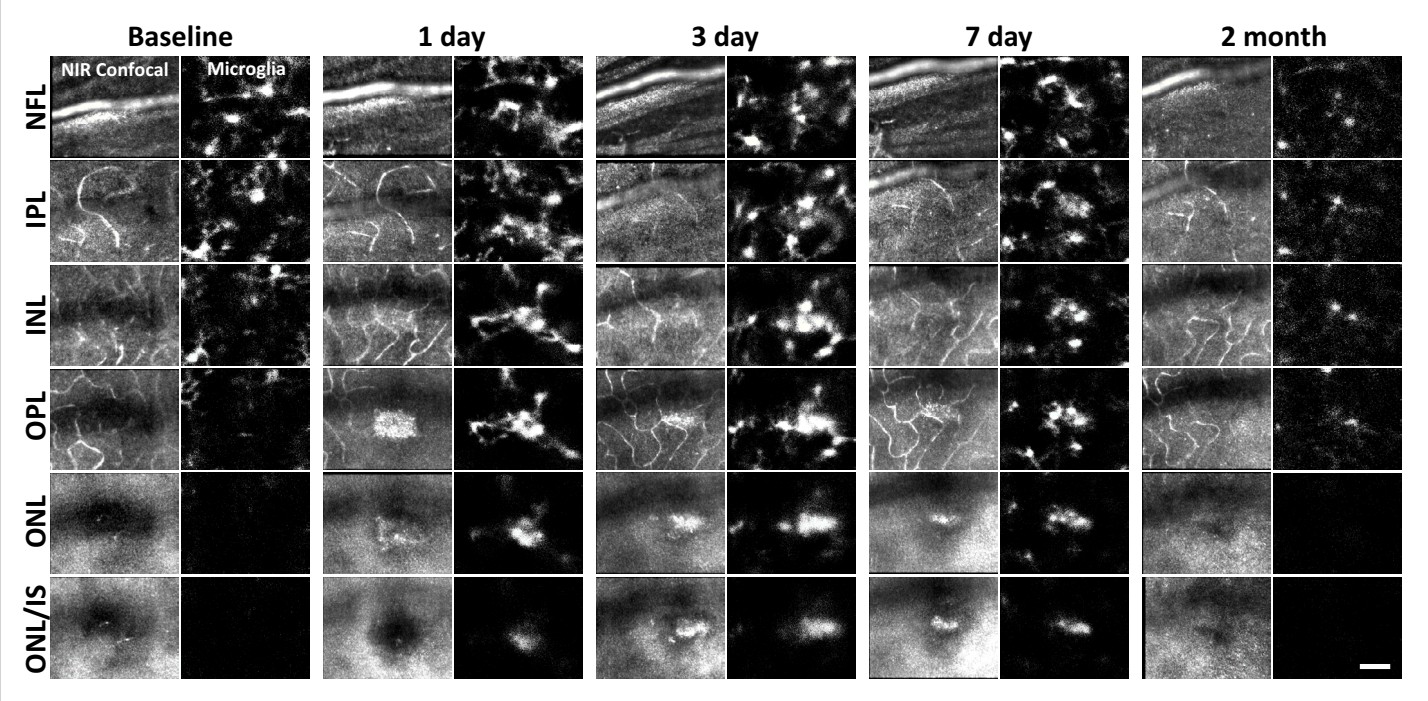

**Figure 6.** Microglial response to laser injury tracked with adaptive optics scanning light ophthalmoscopy (AOSLO). Simultaneously acquired near-infrared (NIR) confocal and fluorescence AOSLO images across different retinal depths. Data are from one CX3CR1-GFP mouse tracked for 2 months. Microglia swarm to hyperreflective locations within 1 day. Microglia maintain an aggregated density for days and resolve by 2 months after damage. Scale bar = 40 μm.

The online version of this article includes the following figure supplement(s) for figure 6:

**Figure supplement 1.** Hyperreflective appearance emerges before microglia swarm to damage location.

## Despite robust microglial involvement, neutrophils do not extravasate

While we observed a robust microglial response to PR damage, there was no evidence of neutrophil involvement at the times we examined (1-, 3-, 7-day, and 2-month follow-up).

In vivo fluorescence imaging allowed us to track neutrophils with AOSLO in Catchup mice (*Hasenberg et al., 2015*). In healthy mice, we observed a sparse population of circulating neutrophils flowing quickly within the largest retinal vessels (*Figure 7—video 1*). We also observed neutrophils moving through single capillary branches (*Figure 7—video 2*). We found that neutrophils show deformation within the small confines of the capillary lumen, often resulting in tube or pill-shaped morphology seen both in vivo and ex vivo (*Figure 7B*, top).

After laser lesion, we found no evidence of neutrophil aggregation or extravasation for any time point assessed (*Figure 8*, *Figure 8—figure supplement 1*). There was also a notable lack of rolling/crawling neutrophils (or any putative leukocyte) in large arterioles or venules surrounding the injury (*Figure 8—video 1*). Neutrophils closest to the ONL lesion were occasionally detected within the deepest retinal capillaries. However, these neutrophils stayed within the retinal vasculature, as evidenced by their pill-shaped morphology and passage routes that follow the known vascular paths seen in confocal/phase-contrast images (*Figure 8—video 2*). Leukocytes, including neutrophils, often impede flow due to their large size (13.7 μm; *Kornmann et al., 2015*) as they compress through capillaries <7 μm in diameter (*Dholakia et al., 2022*). However, we found no evidence of permanently stopped flow as rare stalls would re-perfuse similar to those previously characterized in healthy mice (*Figure 7A*, *Figure 8—video 3*; *Dholakia et al., 2022*).

As a positive control, we used the endotoxin-induced uveitis (EIU) model to show that we could image extravasated neutrophils. This model is known to induce a strong neutrophil response. With fluorescence AOSLO (*Figure 7—video 3*) and ex vivo confocal microscopy (*Figure 7—figure supplement 1*), we found an abundance of neutrophils within the retinal parenchyma 1 day after LPS injection. This confirms the validity of our experimental paradigm and that extravasated neutrophils can be

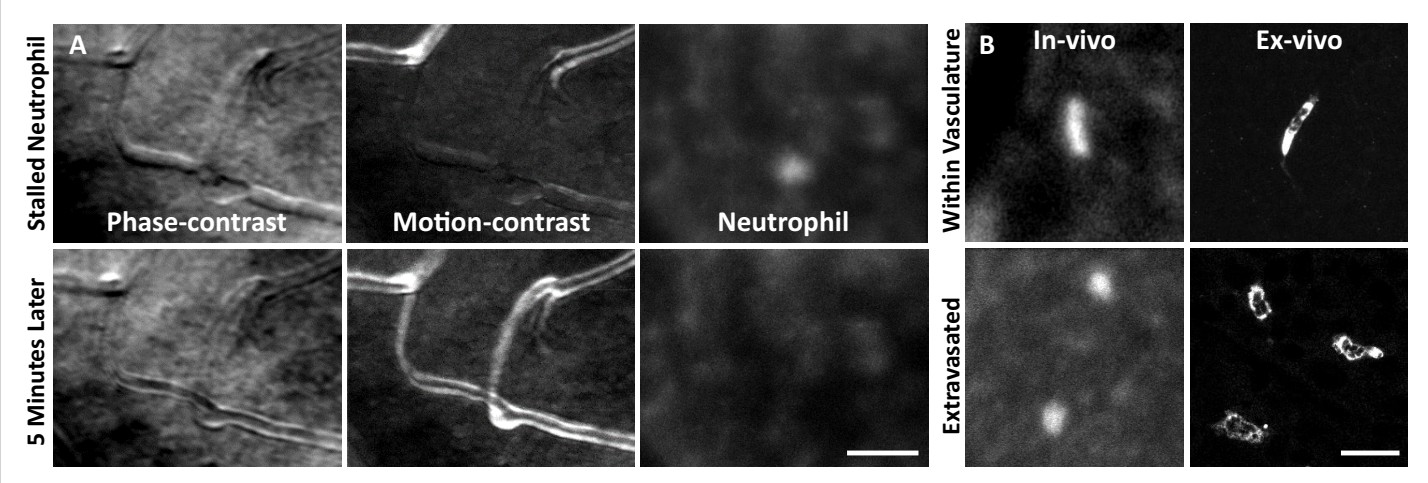

**Figure 7.** Neutrophil morphology imaged in vivo using adaptive optics scanning light ophthalmoscopy (AOSLO). (**A**) Phase-contrast, motion-contrast, and fluorescence AOSLO reveal the impact of passing neutrophils on single capillaries. A rare and exemplary event shows a neutrophil transiently impeding capillary blood flow for minutes in healthy retina. Scale bar = 40 µm. (**B**) In vivo AOSLO and ex vivo fluorescence microscopy show neutrophils in two states. Neutrophils within capillaries displayed elongated, tubular morphology. Extravasated neutrophils were more spherical. Bottom images show extravasated neutrophils in response to an endotoxin-induced uveitis (EIU) model for comparison (not laser damage model). Scale bar = 20 µm.

The online version of this article includes the following video and figure supplement(s) for figure 7:

**Figure supplement 1.** Positive control endotoxin-induced uveitis (EIU) model: wide-field image of ex-vivo neutrophils 1 day post-lipopolysaccharide (LPS) injection.

**Figure 7—video 1.** Neutrophil dynamics within a primary retinal vessel in a healthy Catchup mouse.
https://elifesciences.org/articles/98662/figures#fig7video1

**Figure 7—video 2.** Neutrophils imaged in capillaries of a healthy Catchup mouse.
https://elifesciences.org/articles/98662/figures#fig7video2

**Figure 7—video 3.** Positive control endotoxin-induced uveitis (EIU) model: neutrophils imaged 1 day post-lipopolysaccharide injection.
https://elifesciences.org/articles/98662/figures#fig7video3

imaged with these modalities. Moreover, we found that neutrophils that have extravasated into the retinal parenchyma tended to have a more spherical morphology rather than the compressed, pill-shaped morphology of neutrophils within capillaries (*Figure 7b*).

## Ex vivo analysis confirms in vivo findings

To confirm our in vivo findings, we examined fluorescent microglia and neutrophils in laser-damaged retinal whole mounts imaged with confocal microscopy. The progressive nature of the microglial response to PR damage and general lack of neutrophil participation corroborated in vivo findings.

## Ex vivo: Microglia display dynamic morphological changes in lesion areas

Without laser damage, microglia exhibited a tiled distribution and stellate morphology with highly ramified branching patterns (*Figure 9A*). 1 day after laser exposure, microglial somas aggregated to the lesion location. They began to migrate into the ONL and changed from the ramified morphology seen in the healthy retina, to a dagger-like axial morphology (*Figure 9B*). At 3 and 7 days after lesion, microglia migrated deeper into the outer retina and remained aggregated with an axially elongated phenotype. 2 months after lesion, microglial somas were no longer found in the ONL. Instead, microglia redistributed similar to that of the healthy retina, once again co-stratifying predominantly with the NFL and plexiform layers of the retina (*Figure 9A and B*, *Video 1*).

Whereas we found a robust response of deep microglia at the OPL to PR injury, we observed little migratory response of the microglia in the NFL or IPL (*Figure 9*, *Figure 9—figure supplement 1*), suggesting axial and lateral constraints on the extent of microglial recruitment.

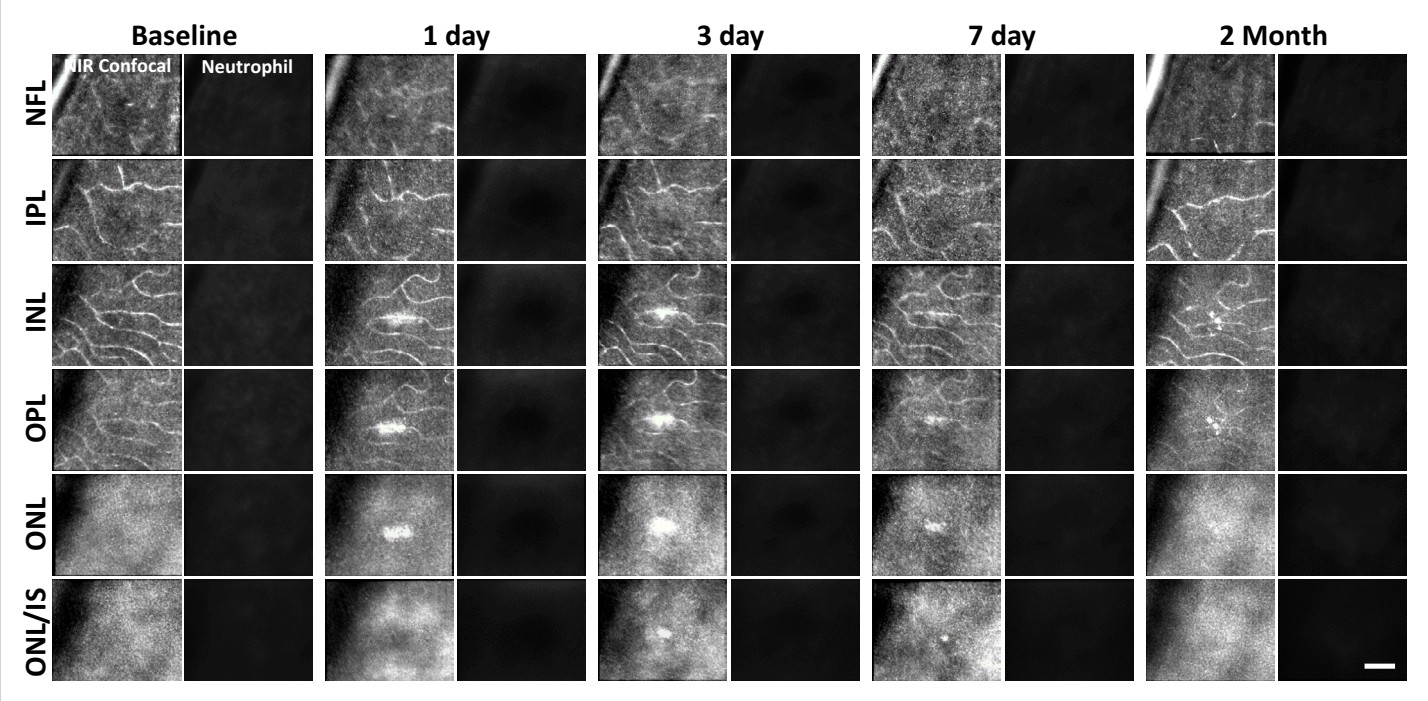

**Figure 8.** Neutrophil response to laser injury tracked with adaptive optics scanning light ophthalmoscopy (AOSLO). A single retinal location was tracked in a Catchup mouse from baseline to 2 months after lesion. Location of the lesion is apparent at 1 and 3 days post-injury with diminishing visibility after 1 week. We did not observe stalled, aggregated, or an accumulation of neutrophils at any time point. This evaluation was confirmed at multiple depths ranging from the nerve fiber layer (NFL) to the outer nuclear layer (ONL). Scale bar = 40 µm.

The online version of this article includes the following video and figure supplement(s) for figure 8:

**Figure supplement 1.** Acute neutrophil response to laser injury tracked with adaptive optics scanning light ophthalmoscopy (AOSLO).

**Figure 8—video 1.** Neutrophil dynamics within a primary retinal vessel in a Catchup mouse 1 day after a deep retinal laser lesion is placed nearby. https://elifesciences.org/articles/98662/figures#fig8video1

**Figure 8—video 2.** Neutrophil dynamics within an outer plexiform layer capillary in a Catchup mouse 1 day after deep laser injury. https://elifesciences.org/articles/98662/figures#fig8video2

**Figure 8—video 3.** Neutrophil dynamics 2.5 hours post-lesion. https://elifesciences.org/articles/98662/figures#fig8video3

## Microglia form PR-containing phagosomes

A detailed examination of microglial involvement with PR somata in response to PR injury revealed unique microglia–PR interactions within the ONL. In cross-section, PR cells comprise a large portion of the retina reaching from the outer-segment tip, inner segments, somata, and spherule/pedicle synaptic contacts at the OPL. Likewise, we saw microglial involvement with all of these layers. Within 1 day, we found microglial processes interspersed within the dense aggregation of PR somata within the ONL. Amoeboid cells enveloped the somata of PRs, and phagocytosis was detected 1, 3, and 7 days post-laser exposure (*Figure 9—figure supplement 2*). Confocal microscopy revealed GFP-positive processes surrounding PR nuclei, engulfing multiple somata (*Figure 9—figure supplement 2A and C*). By 3 days, microglial processes and somas migrated deeper, with processes extending into the distal portions of the PR cells including the inner/outer segments. By 2 months, microglial processes and somas had retreated out of the ONL.

DAPI nuclear stain combined with confocal microscopy not only helped us discern retinal neurons, but also allowed us to differentiate between microglia and phagocytosed PRs. Two features were different from PR and microglial nuclei. First, each PR displayed a uniform, homogeneous nuclear fluorescence, while microglial nuclei appeared heterogeneous and mottled (*Figure 9—figure supplement 2A*). Second, microglial nuclei were nearly 3× larger in volume compared to PR nuclei. Microglia had a nuclear volume of 110±42 µm (*Boycott and Hopkins, 1981*) and PR nuclei were 35±5 µm

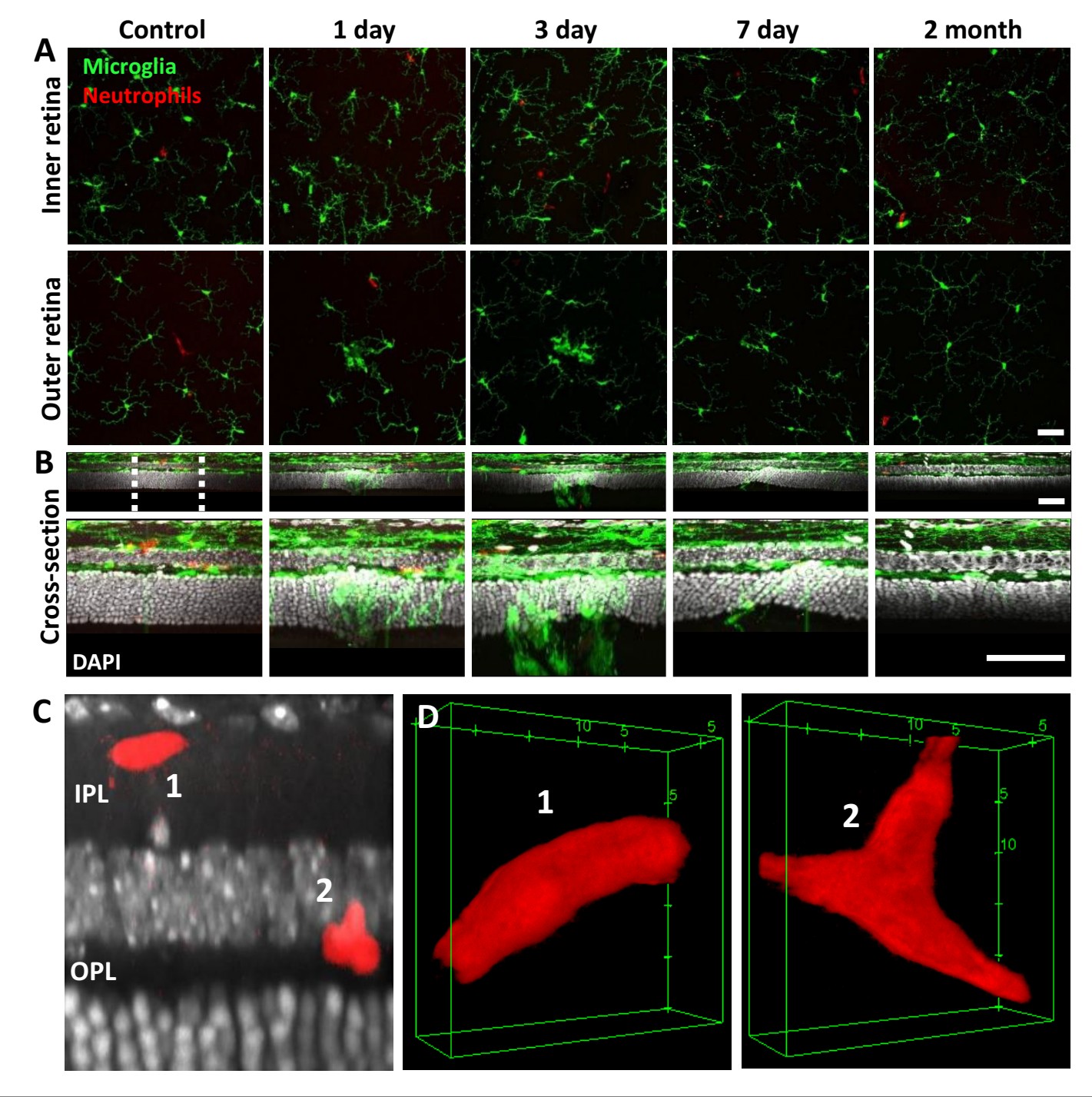

**Figure 9.** Neutrophil and microglial behavior after laser injury, as observed through ex vivo confocal microscopy. (**A**) En-face max intensity projection images of inner and outer (separated by approximate inner nuclear layer [INL] center) retinal microglia/neutrophils in Ly-6G-647-stained CX3CR1-GFP retinas. Microglia display focal aggregation in the outer retina for 1-, 3-, and 7-day time points that is resolved by 2 months. Neutrophils do not aggregate or colocalize to the injury location at any time point. Z-stacks were collected from five mice for the indicated time points. (**B**) Cross-sectional views of en-face z-stacks presented in (**A**), including DAPI nuclear label. White dotted line indicates 100 μm region expanded below. Microglia migrate into the outer nuclear layer (ONL) by 1, 3, and 7 days post-laser injury and return to an axial distribution similar to that of control by 2 months. The few neutrophils detected remained within the inner retina. Scale bars = 40 μm. (**C**) Orthogonal view of DAPI-stained retina with Ly-6G-647-labeled overlay 1 day post-laser-injury. In a rare example, two neutrophils are found within the inner plexiform layer (IPL)/outer plexiform layer (OPL) layers despite a nearby outer retinal laser lesion. Scale bar = 20 μm. (**D**) Magnified 3D cubes representing cells 1 and 2 in (**C**). Cell 1 displays pill-shaped morphology, and cell 2 is localized to a putative capillary branch point. Each is confined within vessels suggesting they do not extravasate in response to laser injury.

*Figure 9 continued on next page*

*Figure 9 continued*

The online version of this article includes the following figure supplement(s) for figure 9:

**Figure supplement 1.** Neutrophil/microglial response to laser injury tracked with ex vivo confocal microscopy.

**Figure supplement 2.** Microglial photoreceptor (PR) phagosomes in the outer retina assessed with ex vivo confocal imaging.

(*Boycott and Hopkins, 1981*) (p<0.001, *Figure 9—figure supplement 2B*). PR nuclei within microglial phagosomes displayed similar nuclear volume compared to adjacent PRs in undamaged locations (*Figure 9—figure supplement 2a*).

## Ex vivo: At lesion sites, neutrophils remain within the retinal vasculature

Corroborating in vivo AOSLO findings, we did not find neutrophil aggregation or extravasation in response to laser damage at any of the time points examined. Ex vivo microscopy revealed that neutrophils were found within the vascular network and did not extravasate into the neural retina. Very few neutrophils were found in lesioned retinas, comparable to healthy retinas. The few detected neutrophils were remnants of those found within the vascular network at the point of death. Detailed 60× Z-stacks revealed 'pill-shaped' morphology similar to that seen in vivo, where neutrophils are compressed within capillaries. This morphology contrasts with the spherical shape of neutrophils in response to the EIU positive-control model (*Video 2*). In one exceptional example (1 day post-damage), we observed two neutrophils in a single z-stack, one in a capillary (1) and one at a capillary branch point (2), but none were observed to have left the confines of the vasculature, suggesting they were not recruited by activated microglia (*Figure 9C and D*, *Video 2*).

Retinal neutrophil concentrations were quantified from larger z-stacks (796 × 796 µm, *Figure 10A*). Control locations (n=2 mice, four z-stacks) had 15±8 neutrophils per mm$^2$ of retina, whereas lesioned locations (n=2 mice, four z-stacks) had 23±5 neutrophils per mm$^2$ of retina (*Figure 10B*). The difference between control and lesioned groups was not statistically significant (p=0.19).

Finally, when we rendered confocal Z-stacks in cross-sectional view, we discovered that the few fluorescent neutrophils present were found colocalized within the tri-layered vascular stratifications of the mouse retina (*Schallek et al., 2013*; *Paques et al., 2003*; *Stahl et al., 2010*). None were found in nonvascular layers. Taken together, all ex vivo data indicates that regardless of the robust phagocytic microglial response, neutrophils remain localized within vessels, suggesting that neutrophils do not extravasate or aggregate in response to this laser lesion model. These data corroborate the in vivo findings.

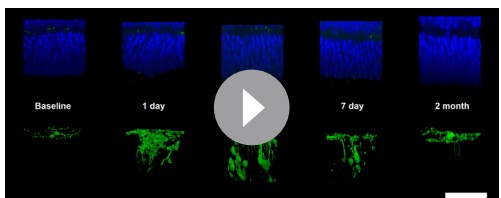

**Video 1.** Rotating 3D cubes of outer retinal nuclei and microglia after focal laser injury. Outer retinal Z-stacks of DAPI-stained whole-mount CX3CR1-GFP retinal tissue were imaged for control, 1-, 3-, 7-day and 2-month time points (n=5 mice). DAPI+ microglia composite cubes are displayed above and microglia-only cubes are displayed below. By 1 day, microglia send projections into the outer nuclear layer (ONL); by 3 and 7 days, microglial somas have migrated into the ONL. Microglia within the ONL are less ramified compared to the baseline condition. By 2 months, microglia are found back within the outer plexiform layer, exhibiting lateral projections, similar to baseline. Scale bar = 40 µm.

https://elifesciences.org/articles/98662/figures#video1

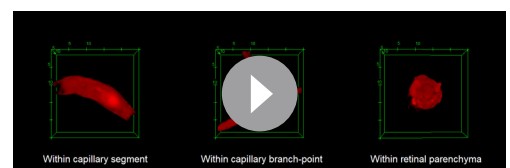

**Video 2.** Rotating 3D cubes of single neutrophils after laser injury or EIU. Ex vivo confocal z-stacks (0.1 µm steps) allowed detailed visualization of single neutrophils 1 day after laser injury or 1 day after intravitreal lipopolysaccharide (LPS) injection. After laser injury, neutrophils maintain a tubular, pill-shaped morphology (left). Occasionally, they would come to rest at capillary branch points (middle). In the EIU model, neutrophils extravasate into the retinal parenchyma and exhibit more spheroid morphology (right). We did not observe neutrophils to exhibit the extravasated morphology in response to laser injury.

https://elifesciences.org/articles/98662/figures#video2

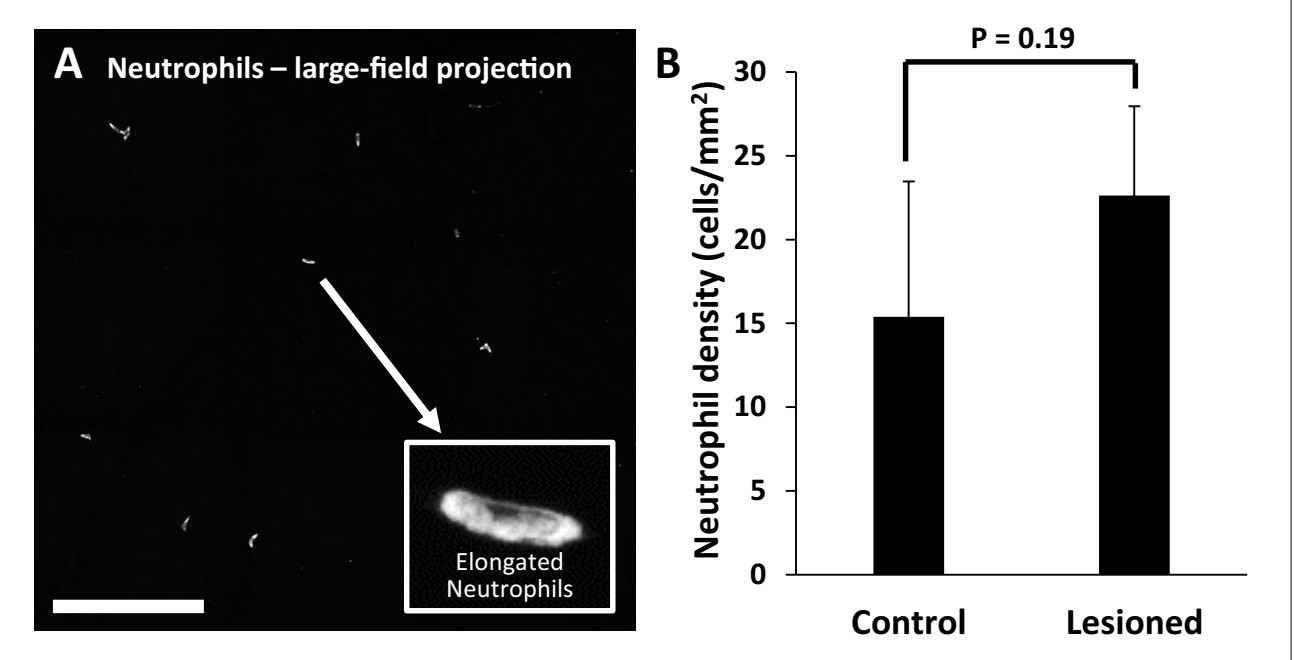

**Figure 10.** Quantification of neutrophils in laser-damaged retinas assessed with ex vivo confocal microscopy over a wide field. (**A**) Representative image (maximum intensity projection) displays neutrophils quantified using large-field (796×796 µm) z-stacks for control or 1 day after injury time points. In both control and laser-injured retinas, neutrophils were sparse and confined to locations within capillaries, suggesting they were the native fraction of circulating neutrophils at time of death. Inset displays an expanded image of a single neutrophil. Scale bar = 200 µm. (**B**) Neutrophils quantified and displayed as the number of neutrophils per retinal area. The difference in the number of neutrophils in control (n=4 locations, 2 mice) vs lesioned (n=4 locations, 2 mice) retinas was not statistically significant (p=0.19, student's paired two-tailed *t*-test). Error bars display mean ± 1 SD.

## Discussion

### Summary

In many retinal diseases, resident microglial populations are found to exhibit cross-talk with systemic immune cells (*Okunuki et al., 2019*; *Boyce et al., 2022*). The complex temporal dynamics between resident and systemic immune cells unfolds from seconds to months and has been poorly characterized due to lack of resolution, sufficient contrast, and a non-destructive imaging approach that can track changes over time. Here, we overcome these limitations by using phase-contrast and fluorescent imaging with adaptive optics to visualize the interaction of multiple cell types. With this advanced imaging technology, we reveal the absence of neutrophil involvement in response to an acute injury, despite progressive changes in microglial activity and morphology. Here we discuss the implications of such findings in the context of retinal damage and more broadly, retinal disease.

### Focal retinal lesions for tracking the immune response

Seminal studies in the brain have used focal light exposure to induce targeted, acute microglial responses (*Nimmerjahn et al., 2005*; *Davalos et al., 2005*). Since then, it has been a popular damage model. Focal lesions in the retina have been conducted in a number of popular animal models including mouse (*Shah et al., 2015*; *Khan et al., 2023*), macaque (*Strazzeri et al., 2014*; *Dhakal et al., 2020*; *Schwarz et al., 2018*), and cats (*Calford et al., 1999*; *Dreher et al., 2001*). Laser damage is also clinically relevant as it is experienced with accidental laser exposure (*Vitellas et al., 2022*; *Linton et al., 2019*), as well as purposeful ablation used in photocoagulation and as a therapy for retinal ischemic disease, retinopathy of prematurity (*American Academy of Ophthalmology, 2014*), and diabetic retinopathy (*Reddy and Husain, 2018*). Thus, our results shed light toward the immune response that may be imparted due to phototoxic damage.

Light damage offers a number of advantages for creating acute damage in the retina. It offers placement precision that is better controlled than other invasive methods, such as incision/poke injuries (*Senut et al., 2004*), retinal detachment (*Maidana et al., 2021*), and chemical injury (*Wan et al.,*

*2008*; *Rösch et al., 2015*). In the mouse, focal light dosage comes with greater axial confinement within the retina by nature of the high numerical aperture of the mouse eye (*Geng et al., 2012*), which imposes minimal collateral damage to layers above or below the plane of focus (*Figure 1A*). The damage took on an elliptical form, likely due to (1) eye motion from respiration and heart rate which spreads the light over a larger integrative area (rather than line). (2) The impact of focal light scatter. (3) A micron-thin line imparting damage on cells that are many microns across manifesting as an ellipse. Most light exposures produced lesions of this elliptical shape. For the reasons described above, some exposures failed to produce a strong, focal damage phenotype. To improve lesion reproducibility, future experiments should better control for subtle eye motions affecting light distribution, especially for long exposures.

## Titration of the 488 nm laser leads to mild retinal damage

While it is beyond the scope of this report to catalog the vast parameter space in which light may impose damage in the retina, it is noteworthy to discuss that the exposure intensity, subtended angle, and duration of the light source we used delivered an intentionally mild damage to the retina. Using dosages too high would run the risk of collateral damage to the blood–retinal barrier or Bruch's membrane (*Haupert et al., 2003*), which could cause a choroidal neovascularization phenotype (*Shah et al., 2015*). This would be undesirable for studying the native/systemic immune response as it could confound the traditional extravasation pathways of the inflammatory system (*Joseph et al., 2020*; *Ley et al., 2007*). Instead, we chose our laser exposure condition to impart a weak, but reproducible loss of PRs while minimizing damage to the cells above and below the plane of focus.

## Progressive PR loss and morphology of the lesion tracked over time

The amount of light delivered did not immediately ablate PRs as peak cell loss was not observed until 3–7 days after laser exposure, suggesting a progressive damage phenotype of apoptosis or necroptotic death. Histology showed ~27% reduction in PR nuclei 7 days post-lesion. The total number of PR nuclei within the lesion location returned to values similar to baseline by 2 months, which was surprising given that PRs do not regenerate. Our interpretation is that adjacent PRs spill into the region of loss, similar to what others have described previously (*Strazzeri et al., 2014*; *Busch et al., 1999*). Corroborating the histological finding of mild cell loss, OCT shows no evidence of local edema, cavitation, or excavation of the ONL after PR loss (*Figure 2A*). The return of the appearance, thickness, and redistribution of cells within the ablated location indicates that retinal remodeling occurs within 2 months. The concurrence of microglia in these outer retinal bands (where they are normally absent; *Lee et al., 2008*) supports the hypothesis that microglia localized to this region facilitate a phagocytic injury response and may contribute toward synaptic remodeling (*Tremblay and Majewska, 2011*).

## CX3CR1-GFP mice exhibit fluorescence not only in microglia

We recognize that the CX3CR1-GFP model can also label systemic cells such as monocytes/macrophages (*Jung et al., 2000*). While it is possible these cells could infiltrate the retina in response to the lesion, we find it unlikely since there was no indication of the leukocyte extravasation cascade (rolling/crawling/stalled cells) within the nearest retinal vasculature. In addition to microglia, retinal perivascular macrophages and hyalocytes also exhibit GFP fluorescence, and these cells may also contribute toward damage resolution.

## The hyperreflective phenotype does not arise from microglia or neutrophils

Light damage is known to create hyper-reflective bands in OCT imaging (*Dhakal et al., 2020*; *Miller et al., 2019*; *Wu et al., 2018*). A common speculation is that the increased backscatter may arise from local inflammatory cells that activate or move into the damage location. In our data, confocal AOSLO and OCT revealed a hyperreflective band at the OPL/ONL after 488 nm light exposure (*Figure 2A and B*). We found that the hyperreflective bands appeared within 30 minutes after the laser injury, preceding any detectable microglial migration toward the damage location (*Figure 6—figure supplement 1*). We thus conclude that the initial hyperreflective phenotype is not caused by microglial cell activity or aggregation.

## Direct activation of microglia from 488 nm light exposure was minimal

It is conceivable that 488 nm light used for either imaging (56 µW) or imparting damage (785 µW) might activate the GFP-containing microglia used here. However, several lines of evidence speak against this possibility. (1) We did not observe photobleaching of CX3CR1-positive cells in response to the damage, suggesting the light was insufficient to damage microglia. (2) For retinal injury, the focal plane was adjusted such that the dose was axially concentrated onto the outer retina. Thus, the light dosage received by microglia above was defocused and less than the targeted PR layer (*Figure 1A*). (3) Histology showed no evident necrotic/apoptotic microglial morphologies.

## 488 nm laser lesion does not photocoagulate or alter retinal circulation

Despite imparting damage to the PRs, the damage regime used here did not alter the perfusion of the retinal circulation. We show three independent measures that blood flow is uninterrupted, despite PR loss and activation of microglia. (1) Fluorescein angiography (*Figure 1B*, bottom) revealed an absence of vascular leakage. (2) AOSLO motion-contrast vascular maps (*Sulai et al., 2014*; *Guevara-Torres et al., 2016*; *Schallek et al., 2013*) displayed persistent blood perfusion inside vessels near lesion sites (*Figure 4*). (3) Capillary line scans indicate that RBC flux was not modified at lesion locations and fell within the normal range (*Guevara-Torres et al., 2016*; *Dholakia et al., 2022*; *Figure 4—figure supplement 1*). Altogether, these three lines of evidence indicate that the lesion did not compromise the blood–retinal barrier or impart perfusion changes within the retinal vasculature.

## Resident microglia do not need systemic neutrophils for resolution of mild laser damage

The CNS and retina, unlike other peripheral tissues, cannot suffer from excess inflammation as there may be dire functional consequences. Therefore, it is possible that microglia protect against exorbitant inflammation by modulating the recruitment of systemic inflammatory cells (*Silverman and Wong, 2018*; *Okunuki et al., 2019*; *Kremlev et al., 2004*; *Babcock et al., 2003*). Of these, neutrophils are often one of the first systemic responders. Despite their helpful roles in other tissues, neutrophils can secrete neurotoxic compounds that could present a danger to the CNS (*Allen et al., 2012*). Given their conflicted role in the body, we ask the question: to what extent do neutrophils respond to acute neural loss in the retina? Retinal cells are lost with age (*Panda-Jonas et al., 1995*) and disease (*Margalit and Sadda, 2003*) and yet, for the organism, visual perception must persist. There are limitations on how generalizable this mild damage is to other damage or disease phenotypes, but this acute damage model can provide clues about how immune cells interact in response to PR loss. In this laser lesion model, we ablate 27% of the PRs in a 50 µm region.

We find that microglia undergo a rapid and progressive response to this injury. We show evidence of PR phagocytosis (*Figure 9—figure supplement 2*), interaction with neighboring microglia (*Figure 9—figure supplement 1*), and they are also axially positioned to facilitate retinal remodeling (*Figure 9*). Furthermore, throughout the temporal evolution of the microglial response, we find no evidence of neutrophil recruitment despite the damage being within 10s of microns from retinal vessels that carry them. At the onset of the neutrophil extravasation cascade, endothelial cells in the vicinity of inflamed tissue typically elevate the expression of adhesion molecules, facilitating the adherence and extravasation of circulating neutrophils (*Sadik et al., 2011*). Furthermore, neutrophils are dependent on priming events as prerequisite to further activation and engagement of their effector functions (*Worthen et al., 1987*; *Linas et al., 1992*). Based on our data, we suggest that although microglia show a strong and lasting activation, at no time point from seconds to months are the damage-associated molecular patterns or chemotactic gradients strong enough to recruit neutrophils in response to this damage. This is evidenced in our data from two key observations: (1) we saw no examples of systemic leukocytes rolling in vessels adjacent to injury locations (*Figure 8—video 1*). (2) We did not observe adherent or extravasated neutrophils adjacent to imparted PR loss (*Figure 8*). This may suggest that the region of insult is too small, or that activated microglia are not sufficient for recruiting neutrophils with damage of this magnitude. Perhaps a minimum threshold of neural damage must be met before neutrophils will respond. It is possible that resident microglia facilitate the necessary phagocytic and retinal remodeling response despite release of cytokines from damaged retinal cells that would normally recruit systemic immune cells in peripheral tissues. Such a strategy would benefit the CNS.

Future work will explore whether there is a threshold magnitude of neuronal cell loss required for recruitment of systemic cells that is unique to the retina. Next studies will examine more severe or widespread injury regimes that provide stronger activating molecular signals and interact with a larger population of systemic cells.

## Microglia may inhibit neutrophil activation

Microglia may be involved in a system that protects the CNS from propagating a larger systemic response, potentially exacerbating disease pathologies that would compromise overall CNS function. In another damage model (*Uderhardt et al., 2019*), they report that tissue-resident macrophages may exhibit the capacity to cloak tissue micro-damage. This offers the possibility that resident immune cells, such as retinal microglia, can handle small insults without inducing a chemokine cascade that may invoke a larger systemic response that could further damage the precious retinal tissue (*Pfeifer et al., 2023*). Regardless of the mechanism, we find that despite a robust microglial activation that lasts for weeks, at no time point do they recruit neutrophils. The nuance of this interaction likely represents the fine balance that facilitates a helpful local response within the CNS that does not impart a widespread cytokine storm that may otherwise exacerbate retinal damage. Further work will explore whether microglia exhibit a cloaking response in the retina, inhibiting neutrophil or other immune cell extravasation/chemotaxis toward lesion sites. We expect such work to be pivotal in understanding the balance that is broken or left unchecked in conditions of autoimmune disease and the umbrella of diseases that comprise the uveitic response, a direct threat to lifelong vision (*Miserocchi et al., 2013*).

## Conclusion

Here, we have applied innovative in vivo imaging at the microscopic scale to reveal the cellular immune response to a retina in jeopardy. The dynamic environment of the retina includes a native population of resident microglia and systemic immune cells delivered through the vasculature. These two lines of defense work in concert in the mammalian body and are critical for maintaining retinal homeostasis. In this work, we directly study the interaction between microglia and neutrophils, two major classes of immune cells that are implicated in inflammatory initiation, escalation, propagation, and debris removal in response to acute geographical injury in the retina. Using cutting-edge retinal imaging modalities, we find that resident microglia become locally activated and regionally responsive to focal laser lesion. They migrate away from their stratified locations near plexiform layers of the retina and toward the site of damage within hours to weeks after injury. However, systemic neutrophils, which are typically regarded as first-line responders to tissue damage, are not recruited to this damage despite neutrophils flowing within 10s of microns away from the location of damage (*Figure 8—video 2*). Beyond the context of this specific finding, we share this work with the excitement that AOSLO cellular-level imaging may reveal the interaction of multiple immune cell types in the living retina. By using fluorophores associated with specific immune cell populations, the complex dynamics that orchestrate the immune response may be examined in this specialized tissue. This work and future studies may reveal further insights to the interactions of single immune cells in the living body in a noninvasive way.

## Limitations and future directions

This work represents some of the earliest reports of single immune cell interactions in the living retina. To narrow the scope, we focus on one type of injury, a targeted elimination of photoreceptors. Thus, these findings are limited to one type of injury that may be experienced in the retina. We expect that these seminal demonstrations will serve as a platform for future studies that examine the large parameter space of retinal damage. These include (1) conditions of greater severity with increased power, duration, and extent of light damage. (2) Models of systemic and local infection. (3) Response to therapy that may modulate the immune response. (4) Examine immune cell activity in models of retinal disease such as diabetic retinopathy, glaucoma, and age-related macular degeneration, each expected to reveal the nuance of the coordinated immune response.

# Materials and methods

**Key resources table**

| Reagent type (species) or resource | Designation | Source or reference | Identifiers | Additional information |
|---|---|---|---|---|
| Strain, strain background (*Mus musculus*) | C57BL/6J | The Jackson Laboratory | Strain #: 000664 RRID:IMSR_JAX:000664 | |
| Strain, strain background (*M. musculus*) | CX3CR1-GFP | The Jackson Laboratory | Strain #: 005582 RRID:IMSR_JAX:005582 | |
| Strain, strain background (*M. musculus*) | Catchup | Laboratory of M. Gunzer | C57BL/6-*Ly6g*(tm2621CretdTomato)Arte | *Hasenberg et al., 2015* |
| Antibody | Ly-6G-647 | BioLegend | Cat#: 127610; RRID:AB_1134159 | IF (1:200) |
| Other | DAPI stain | Cell Signaling Technology | Cat#: 4083 | 1:500 of 10 mg/ml stock |
| Chemical compound, drug | Lipopolysaccharide (LPS) | Sigma-Aldrich | Cat#: L4391 | 1 ng (1 µl) delivered intravitreally |
| Software, algorithm | Cell Counter plugin | ImageJ FIJI | Cell Counter plugin | Author: Kurt De Vos, ImageJ version 1.53q |
| Other | AK-FLUOR Fluorescein Sodium 10% | McKesson | Cat#: 1120803 | ~100 µl injection of 10mg/ml stock |

## Mice

All experiments herein were approved by the University Committee on Animal Resources (protocol #: UCAR-2010-052E) and according to the Association for Research in Vision and Ophthalmology statement for the Use of Animals in Ophthalmic and Vision Research as well as institutional approvals by the University of Rochester. C57BL/6J (#000664, Jackson Labs, Bar Harbor, ME) mice were used to track the retinal phenotype after laser exposure. Heterozygous CX3CR1-GFP (#005582, Jackson Labs) mice were used to track GFP-expressing microglia. Mice with heterozygous transgenic expression of tdTomato in neutrophils ('Catchup' mice) were provided by the foundry lab of M. Gunzer (*Hasenberg et al., 2015*). 18 mice (10 males, 8 females) in total were used for in vivo imaging. 10 additional mice (5 males, 5 females) were used for ex vivo histology. Ages for all mice used for this work were postnatal weeks 6–24.

## Preparation for in vivo imaging

Mice were anesthetized with intraperitoneal injection of ketamine (100 mg/kg) and xylazine (10 mg/kg). The pupil was dilated with 1% tropicamide (Sandoz, Basel, Switzerland) and 2.5% phenylephrine (Akorn, Lake Forest, IL). A custom contact lens (1.5 mm base curve, 3.2 mm diameter,+10 diopter power, Advanced Vision Technologies, Lakewood, CO) was fitted to the eye. For a subset of experiments, 50 mg/kg fluorescein (AK-FLUOR, Akorn, Decatur, IL) was administered by intraperitoneal injection to confirm vascular integrity and perfusion status after injury. During AOSLO imaging, anesthesia was supplemented with 1% (v/v) isoflurane in oxygen and mice were maintained at 37°C via electric heat pad. Eye hydration was maintained throughout imaging with regular application of saline eye drops (Refresh tears, Allergan, Sydney, Australia) and lubricating eye gel (Genteal, Alcon Laboratories Inc, Fort Worth, TX). Mice were placed in a positioning frame with 6 degrees of freedom to allow for stable animal positioning and aid in retinal navigation.

## In vivo AOSLO imaging

Four light sources were used for AOSLO imaging. A 904 nm diode source (12 µW, Qphotonics, Ann Arbor, MI) was used for wavefront sensing. A second 796 nm superluminescent diode (196 µW, Superlum, Cork, Ireland) was used for reflectance imaging, including confocal and phase-contrast modes (*Guevara-Torres et al., 2016*; *Geng et al., 2012*). A third 488 nm light source (56 µW, Toptica Photonics, Farmington, NY) was used to visualize GFP-positive microglia in CX3CR1-GFP mice. A fourth 561 nm light source (95 µW, Toptica Photonics) was used to visualize tdTomato-positive neutrophils in Catchup mice. All light sources were fiber-coupled and axially combined through the

AOSLO system (*Geng et al., 2012*). Fast (15.4 kHz) and slow (25 Hz) scanners create a raster scan pattern, which is relayed through a series of afocal telescopes to and from the eye. A Shack-Hartmann wavefront sensor (consisting of a lenslet array and a Rolera XR camera, QImaging, Surrey, Canada) measures the aberrations of the eye and a deformable mirror (ALPAO, Montbonnot-Saint-Martin, France) provided the wavefront correction. Reflected 796 nm light was collected with a photomultiplier tube (H7422-50, Hamamatsu Photonics, Hamamatsu, Japan). All confocal reflectance images were captured with a 30 µm pinhole (1.3 Airy Disc Diameters, ADD). Phase contrast was achieved by displacing the pinhole relative to the principal axis of the detection plane as previously described (*Guevara-Torres et al., 2020*). Fluorescence was captured with a photomultiplier tube (H7422-40, Hamamatsu) either coupled with a 520Δ35 band-pass filter (FF01-520/35-25, Semrock, Rochester, NY) for GFP emission or a 630Δ92 band-pass filter (FF01-630/92-25, Semrock) for tdTomato emission. All fluorescent images were captured with a confocal 50 µm pinhole (2.1 ADD). Image field sizes were either 4.98° × 3.95° or 2.39° × 1.94°. NIR and visible imaging channels were made coplanar, compensating for longitudinal chromatic aberration by independently focusing each light source onto the same axial structure in the retina. Through-focus stacks were acquired by sequentially changing the focus from NFL to the PR outer segments by using the defocus term on the custom adaptive optics control software.

As described previously, red blood cell (RBC) imaging was achieved by combining phase-contrast imaging with a strategy to arrest the slow galvanometer scanner and let the resonant scanner project a single 'line' (0.71° scan angle) on the retina. This enabled RBC flux imaging by positioning this line orthogonal to the direction of flow for single capillaries. As blood cells moved through capillaries, they were 'self-scanned' producing images of RBCs in space/time (*Guevara-Torres et al., 2016*).

## PR laser damage model and post-injury time points for imaging

488 nm light (continuous wave laser diode, ±4 nm bandwidth, Toptica Photonics) was used to create an acute laser injury. 785 µW of 488 nm light was projected through the AOSLO and focused onto the PR outer segments (*Figure 1A*) for 3 minutes in a single line on the retina subtending 24 × 1 µm to concentrate the power to a small region. Laser injuries were placed between 5 and 15° from the optic disc. To avoid absorption confounds, we refrained from placing lesions beneath large retinal vessels. For experiments examining neutrophil involvement, lesions were placed <100 µm away from retinal veins and within microns of capillaries to increase the chance of extravasation through these preferred pathways in the retina (*Crane and Liversidge, 2008*; *Xu et al., 2002*; *Xu et al., 2004*). As many as four such lesions were placed per retina. This protocol produced a hyper-reflective phenotype in the >40 locations across 28 mice. In rare cases, the exposure yielded no hyperreflective lesion and was often in mice with high retinal motion, where the light dosage was spread over a larger retinal area. These locations were not included in the in vivo or histological analysis.

Throughout this work, we assessed the effects of the laser injury for the following time points: baseline/control, 1 day (18–28 hours), 3 days, 7 days, and 2 months.

## In vivo SLO and OCT imaging

To confirm global ocular health and changes imparted by the laser damage, a commercial Heidelberg Spectralis system (Heidelberg, Germany) was used to acquire SLO and OCT images. 30° and 55° fields were used for SLO acquisitions. The 30° field was used for OCT acquisitions. For some experiments, fluorescein angiography was captured by imaging the retina within 10 minutes of fluorescein administration (details above). The fluorescence mode of the SLO also enabled wide-field images of GFP positive microglia.

OCT was used to provide detailed information regarding the axial nature of the laser damage. We used a coarse scan area to capture several damage locations in a single field (61 B-scans, 1.02 × 0.85 mm). A dense 3D data cube was also captured (49 B-scans, 513 × 171 µm). 'Follow-up' mode, which allows the HRA software to return to the same retinal location, was used whenever possible. To reveal the cross-sectional profile for each lesion, several adjacent B-scans were spatially averaged (~30 µm).

## Preparation for ex vivo imaging

Mice were euthanized by $CO_2$ asphyxiation followed by cervical dislocation. Within 5 minutes of asphyxiation, eyes were enucleated and placed in 4% paraformaldehyde (PFA, diluted from: #15714S, Electron Microscopy Sciences, Hatfield, Pennsylvania) in 1× phosphate-buffered saline (PBS, #806552, Sigma-Aldrich, St. Louis, MO) for 1 hour at room temperature. Eyes were dissected to remove the cornea, lens, and vitreous. Each eye cup was placed in one well of a 24-well plate containing 0.5 ml of 0.8% PFA and left overnight at 4°C. The retina was separated from the retinal pigmented epithelium/choroid with attention to preserve inner retinal layers 'up' orientation. If not applying antibody, the tissue was directly flat mounted (see below). For antibody staining, the retina was placed in 1× BD perm/wash buffer (#554723, BD Biosciences, Franklin Lakes, NJ) with 5% donkey serum (#D9663, Sigma-Aldrich) diluted in PBS for overnight incubation at room temperature with gentle shaking. The following was performed in the dark. Ly-6G-647 antibody (1:200, #127610, RRID:AB_1134159, BioLegend, San Diego, CA) and DAPI (1:500 of 10 mg/ml stock, #4083, Cell Signaling Technology, Danvers, MA) were diluted in 1× perm/wash buffer and retinas were incubated for 3 days at room temperature with gentle shaking. Retinas were washed with PBS three times over 3 hours. The retina was cut into four radially symmetrical petals, flat-mounted on a glass slide in Vectashield mounting buffer (H-1000-10, Vector Labs, Newark, CA) with a #1.5 cover slip (#260406, Ted Pella Inc, Redding, CA), sealed with nail polish, and stored at 4°C until imaged.

## Ex vivo confocal imaging

Whole-mount retinas were imaged with a Nikon A1 confocal microscope (Melville, NY). DAPI (405 nm ex, 441Δ66 nm em), CX3CR1-GFP (488 nm ex, 525Δ50 nm em), and Catchup/anti-Ly-6G-647 (635 nm ex, 665Δ50 nm em) were simultaneously imaged. Z-stacks (0.1 or 0.5 µm step size) at control or laser-damaged locations were acquired with a ×60 oil objective, producing images that were 295 × 295 µm. Larger (796 × 796 µm) z-stacks were acquired by blending several 60× z-stack acquisitions (3 × 3, 15% overlap) using Nikon NIS Elements software. z-stacks were re-sliced (ImageJ FIJI; *Schindelin et al., 2012*; X-Z dimension) to visualize fluorescence depth profiles.

## Endotoxin-induced uveitis protocol

To serve as a positive control and show evidence of known neutrophil invasion, we adopted the EIU model to confirm that fluorescent Catchup neutrophils could be observed in vivo with AOSLO. This was performed in two mice. The EIU model has been described previously (*Chu et al., 2016*; *Rosenbaum et al., 1980*). Briefly, we performed intravitreal injections of lipopolysaccharide (LPS, #L4391, Sigma-Aldrich). Mice were anesthetized with ketamine and xylazine. A 34-gauge Hamilton needle was used to deliver 1 µl (1 ng) of LPS diluted in PBS into the vitreous posterior to the limbus. 1 day post-LPS injection, mice were either imaged with AOSLO (Catchup mice) or collected for ex vivo histology (Ly-6G-647 stained C57BL/6J mice) to confirm fluorescent neutrophil presence in the neural parenchyma.

## AOSLO image processing

To correct residual motion from heart rate and respiration, AOSLO videos were registered with a custom cross-correlation-based frame registration software (*Dubra and Harvey, 2010*; *Yang et al., 2014*). Motion correction was also applied to simultaneously collected fluorescence videos. After registration, confocal, phase-contrast, and fluorescence AOSLO videos were temporally averaged (250 frames, 10 seconds). Blood perfusion maps were computed by calculating the standard deviation of pixel intensity over 30 seconds (*Schallek et al., 2013*).

## INL + ONL nuclei quantification

Raw DAPI z-stacks were used for manual counting of nuclei (n=10 mice, three regions per time point). Analysis regions were circular, with a 50 µm diameter (corresponding to the size and shape of the damage region seen with confocal AOSLO) and analyzed in depth producing volumetric cylinders through the INL or ONL (*Figure 3D and E*). Nuclei were counted manually using the 'Cell Counter' plugin in ImageJ (author: Kurt De Vos, ImageJ version 1.53q).

## PR+ microglia nuclei volume quantification

A single DAPI-stained z-stack (OPL + ONL) from a CX3CR1-GFP mouse was used to quantify nuclear volume of PRs (n=20) and microglia (n=14). Data was centered at a lesion location 3 days post-laser

exposure. ImageJ was used to manually measure the en-face diameter of nuclei in their short and long axis. We averaged the two measurements and assumed spherical shape for analysis. These measurements aided in the differentiation of PRs, invading microglia and PR phagosomes.

## Quantification of RBC flux in capillaries

Using the capillary line-scan approach (*Guevara-Torres et al., 2016*; *Joseph et al., 2015*; *Dholakia et al., 2020*), we quantified RBC flux within an epoch of 1 second. This spanned several cardiac cycles (*Joseph et al., 2019b*; *Feng et al., 2023*). To improve SNR, space-time images were convolved with a Gaussian spatio-temporal filter (σ=7.5 pixels, 0.33 µm). This strategy did not interfere with spatial resolution as pixels oversample the optical point-spread of the AOSLO by >20× (*Guevara-Torres et al., 2016*). RBC flux was determined by manually marking blood cells using the 'Cell Counter' plugin in ImageJ.

## Neutrophil density quantification

Ly-6G-647-stained retinal tissue was used to manually count neutrophils within montaged 796 × 796 µm z-stacks. Cells were counted at control (n=4 locations, two mice) and lesioned (n=4 locations, two mice) locations using the 'Cell Counter' ImageJ plugin. All values are reported as mean ± SD. There were no data points omitted from any of the analysis reported in this work.

## Acknowledgements

We thank Colin Chu, Justin Elstrott, and Tiffany Heaster for useful intellectual conversations. We also thank Minsoo Kim for generously transferring the Catchup mouse strain.

## Additional information

### Competing interests

Derek Power, Jesse Schallek: financial support from Genentech, Inc. Justin Elstrott: is affiliated with Genentech, Inc.

### Funding

| Funder | Grant reference number | Author |
| --- | --- | --- |
| National Eye Institute | EY028293 | Jesse Schallek |
| National Eye Institute | P30 EY001319 | Jesse Schallek |
| Research to Prevent Blindness | Career Advancement Award | Jesse Schallek |
| Research to Prevent Blindness | Department of Ophthalmology Unrestricted Grant | Jesse Schallek |
| Dana Foundation | David Mahoney Neuroimaging Award | Jesse Schallek |
| Genentech, Inc. | Collaborative Research Grant | Justin Elstrott |

The funders had no role in study design, data collection and interpretation, or the decision to submit the work for publication.

### Author contributions

Derek Power, Data curation, Formal analysis, Supervision, Validation, Visualization, Methodology, Writing – original draft, Writing – review and editing; Justin Elstrott, Conceptualization, Funding acquisition, Writing – review and editing; Jesse Schallek, Conceptualization, Resources, Data curation, Software, Formal analysis, Supervision, Funding acquisition, Validation, Investigation, Visualization, Methodology, Writing – original draft, Writing – review and editing

## Author ORCIDs
Derek Power (ID) https://orcid.org/0000-0002-3362-1589
Justin Elstrott (ID) https://orcid.org/0000-0001-6694-5929
Jesse Schallek (ID) https://orcid.org/0000-0002-6337-4187

## Ethics
All experiments herein were approved by the University Committee on Animal Resources (protocol #: UCAR-2010-052E) and according to the Association for Research in Vision and Ophthalmology statement for the Use of Animals in Ophthalmic and Vision Research as well as institutional approvals by the University of Rochester.

Reviewer #2 (Public review): https://doi.org/10.7554/eLife.98662.4.sa1
Reviewer #3 (Public review): https://doi.org/10.7554/eLife.98662.4.sa2
Author response https://doi.org/10.7554/eLife.98662.4.sa3

## Additional files

### Supplementary files
MDAR checklist

### Data availability
Data is publicly available through Dryad. https://doi.org/10.5061/dryad.w3r228143.

The following dataset was generated:

| Author(s) | Year | Dataset title | Dataset URL | Database and Identifier |
| --- | --- | --- | --- | --- |
| Power D, Elstrott J, Schallek J | 2025 | Data from: Photoreceptor loss does not recruit neutrophils despite strong microglial activation | https://doi.org/10.5061/dryad.w3r228143 | Dryad Digital Repository, 10.5061/dryad.w3r228143 |

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
