## [Editor Report · eLife Assessment]

The study by Power and colleagues is **important** as elucidating the dynamic immune responses to photoreceptor damage in vivo potentiates future work in the field to better understand the disease process. The evidence supporting the authors’ claims is **compelling**.

---

## [Referee Report · Reviewer #2 (Public review)]

Summary:

This study uses in vivo multimodal high-resolution imaging to track how microglia and neutrophils respond to light-induced retinal injury from soon after injury to 2 months post-injury. The in vivo imaging finding was subsequently verified by ex vivo study. The results suggest that despite the highly active microglia at the injury site, neutrophils were not recruited in response to acute light-induced retinal injury.

Strengths:

An extremely thorough examination of the cellular-level immune activity at the injury site. In vivo imaging observations being verified using ex vivo techniques is a strong plus.

---

## [Referee Report · Reviewer #3 (Public review)]

Summary

This work investigated the immune response in the murine retina after focal laser lesions. These lesions are made with close to 2 orders of magnitude lower laser power than the more prevalent choroidal neovascularization model of laser ablation. Histology and OCT together show that the laser insult is localized to the photoreceptors and spares the inner retina, the vasculature and the pigment epithelium. As early as 1-day after injury, a loss of cell bodies in the outer nuclear layer is observed. This is accompanied by strong microglial proliferation to the site of injury in the outer retina where microglia do not typically reside. The injury did not seem to result in the extravasation of neutrophils from the capillary network, constituting one of the main findings of the paper. The demonstrated paradigm of studying the immune response and potentially retinal remodeling in the future in vivo is valuable and would appeal to a broad audience in visual neuroscience.

Strengths

Adaptive optics imaging of murine retina is cutting edge and enables non-destructive visualization of fluorescently labeled cells in the milieu of retinal injury. As may be obvious, this in vivo approach is a benefit for studying fast and dynamic immune processes on a local time scale - minutes and hours, and also for the longer days-to-months follow-up of retinal remodeling as demonstrated in the article. In certain cases, the in vivo findings are corroborated with histology.

The analysis is sound and accompanied by stunning video and static imagery. A few different sets of mouse models are used: (a) two different mouse lines, each with a fluorescent tag for neutrophils and microglia, (b) two different models of inflammation - endotoxin-induced uveitis (EAU) and laser ablation are used to study differences in the immune interaction.

One of the major advances in this article is the development of the laser ablation model for 'mild' retinal damage as an alternative to the more severe neovascularization models. This model would potentially allow for controlling the size, depth and severity of the laser injury opening interesting avenues for future study.

The time-course, 2D and 3D spatial activation pattern of microglial activation are striking and provide an unprecedented view of the retinal response to mild injury.

Editor's note: The authors have addressed all the previous concerns raised by the reviewers.

---

## [Author Response]

The following is the authors’ response to the previous reviews

**Public Reviews:**

**Reviewer #2 (Public review):**
Summary:This study uses in vivo multimodal high-resolution imaging to track how microglia and neutrophils respond to light-induced retinal injury from soon after injury to 2 months post-injury. The in vivo imaging finding was subsequently verified by ex vivo study. The results suggest that despite the highly active microglia at the injury site, neutrophils were not recruited in response to acute light-induced retinal injury.Strengths:An extremely thorough examination of the cellular-level immune activity at the injury site. In vivo imaging observations being verified using ex vivo techniques is a strong plus.

Thank you!

Weaknesses:This paper is extremely long, and in the perspective of this reviewer, needs to be better organized. Update: Modifications have been made throughout, which has made the manuscript easier to follow.

Thank you!

Study weakness: though the finding prompts more questions and future studies, the findings discussed in this paper is potentially important for us to understand how the immune cells respond differently to different severity level of injury. The study also demonstrated an imaging technology which may help us better understand cellular activity in living tissue during earlier time points.

We agree that AOSLO has much to offer and this represents some of the earliest reports of its kind.

Comments on revisions:I appreciate the thorough clarification and re-organization by the authors, and the messages in the manuscript are now more apparent. I recommend also briefly discussing limitations/future improvements in the discussion or conclusion.

We have added a section to the discussion entitled “Limitations and future improvements”, please see lines 665 – 677.

**Reviewer #3 (Public review):**
SummaryThis work investigated the immune response in the murine retina after focal laser lesions. These lesions are made with close to 2 orders of magnitude lower laser power than the more prevalent choroidal neovascularization model of laser ablation. Histology and OCT together show that the laser insult is localized to the photoreceptors and spares the inner retina, the vasculature and the pigment epithelium. As early as 1-day after injury, a loss of cell bodies in the outer nuclear layer is observed. This is accompanied by strong microglial proliferation to the site of injury in the outer retina where microglia do not typically reside. The injury did not seem to result in the extravasation of neutrophils from the capillary network, constituting one of the main findings of the paper. The demonstrated paradigm of studying the immune response and potentially retinal remodeling in the future in vivo is valuable and would appeal to a broad audience in visual neuroscience.StrengthsAdaptive optics imaging of murine retina is cutting edge and enables non-destructive visualization of fluorescently labeled cells in the milieu of retinal injury. As may be obvious, this in vivo approach is a benefit for studying fast and dynamic immune processes on a local time scale - minutes and hours, and also for the longer days-to-months follow-up of retinal remodeling as demonstrated in the article. In certain cases, the in vivo findings are corroborated with histology.

Thank you!

The analysis is sound and accompanied by stunning video and static imagery. A few different sets of mouse models are used, (a) two different mouse lines, each with a fluorescent tag for neutrophils and microglia, (b) two different models of inflammation - endotoxin-induced uveitis (EAU) and laser ablation are used to study differences in the immune interaction.

Thank you!

One of the major advances in this article is the development of the laser ablation model for 'mild' retinal damage as an alternative to the more severe neovascularization models. This model would potentially allow for controlling the size, depth and severity of the laser injury opening interesting avenues for future study.

Thank you!

The time-course, 2D and 3D spatial activation pattern of microglial activation are striking and provide an unprecedented view of the retinal response to mild injury.

We agree that this more complete spatial and temporal evaluation made possible by in vivo imaging is novel.

WeaknessesGeneralization of the (lack of) neutrophil response to photoreceptor loss - there is ample evidence in literature that neutrophils are heavily recruited in response to severe retinal damage that includes photoreceptor loss. Why the same was not observed here in this article remains an open question. One could hypothesize that neutrophil recruitment might indeed occur under conditions that are more in line with the more extreme damage models, for example, with a stronger and global ablation (substantially more photoreceptor loss over a larger area). This parameter space is unwieldy and sufficiently large to address the question conclusively in the current article, i.e. how much photoreceptor loss leads to neutrophil recruitment? By the same token, the strong and general conclusion in the title - Photoreceptor loss does not recruit neutrophils - cannot be made until an exhaustive exploration be made of the same parameter space. A scaling back may help here, to reflect the specific, mild form of laser damage explored here, for instance - Mild photoreceptor loss does not recruit neutrophils despite...

We are striving for clarity and accuracy in our title without adding too many qualifiers. At present, we feel that the title as submitted is consistent and aligned with the central finding of our manuscript. The nuance that the reviewer points to is elaborated in the body of the manuscript and we hope the general readership appreciates the same level of detail as appreciated by reviewer #3.

EIU model - The EIU model was used as a positive control for neutrophil extravasation. Prior work with flow cytometry has shown a substantial increase in neutrophil counts in the EIU model. Yet, in all, the entire article shows exactly 2 examples in vivo and 3 ex vivo (Figure 7) of extravasated neutrophils from the EIU model (n = 2 mice). The general conclusion made about neutrophil recruitment (or lack thereof) is built partly upon this positive control experiment. But these limited examples, especially in the case where literature reports a preponderance of extravasated neutrophils, raise a question on the paradigm(s) used to evaluate this effect in the mild laser damage model.

This is a helpful suggestion. We agree that readers should see more evidence of the positive control. Therefore we have now included two more supplementary files that show that there is a strong neutrophil response to EIU. In Figure 7 – supplementary figure 1, we show many Ly-6G-positive neutrophils in the retina seen with histology at the 24 hour time point. In Figure 7 – video 3, we show massive Catchup-positive neutrophil presence in vivo at 24hrs as well. This aligns with our positive control and also the literature.

Overall, the strengths outweigh the weaknesses, provided the conclusions/interpretations are reconsidered.

With the added clarification about the magnitude of the neutrophil response in EIU, we feel that the conclusions presented in the manuscript as-is are valid and appropriate.

**Recommendations for the authors:**

**Reviewer #3 (Recommendations for the authors):**
The authors are applauded for embracing the reviewers' feedback and making substantial revisions. Some minor comments below:The weakness noted in the public review encourages the authors to reconsider the interpretations drawn based on the results. One would have expected to see far more examples of extravasated neutrophils from the EIU model. That this was not seen weakens the neutrophil recruitment claim substantially. Even without this claim, the methods, laser damage model, time-course and spatial activation pattern of microglial activation are all striking and unprecedented. So, as stated in the public review, the strengths do indeed outweigh the weaknesses once the neutrophil claim is softened.

We address this in the response above. A strong neutrophil response was observed to EIU. This was confirmed with both histology and in vivo imaging.

This was alluded to by Reviewer 1 in the prior review - at times, there is an overemphasis on imaging technology that distracts from the scientific questions. The imaging is undoubtedly cutting-edge but also documented in prior work by the authors. Any efforts to reduce or balance the emphasis would help with the general flow.

Given that these discoveries are made possible partly through new technology, we prefer to keep the details of the innovation in the current manuscript. Given the exceptionally large readership of eLife, we feel some description of the AOSLO imaging is warranted in the manuscript.